# BMP-9 mediates fibroproliferation in fibrodysplasia ossificans progressiva through TGF-β signaling

Chengzhu Zhao [ID][1,2✉], Yoshiko Inada[2], Souta Motoike[2], Daisuke Kamiya [ID][2,3], Kyosuke Hino[2,4] & Makoto Ikeya [ID][2,3✉]

## Abstract

Fibrodysplasia ossificans progressiva (FOP) is a rare genetic disorder presenting with progressive heterotopic ossification (HO) in soft tissues. Early-stage FOP is characterized by recurrent episodes of painful tissue swelling (flare-ups), with numerous proliferation-activated mesenchymal stromal cells (MSCs) subsequently causing HO. However, the mechanisms underlying flare-up progression remain unclear. In this study, we evaluated the proliferation of MSCs obtained from FOP patient-derived induced pluripotent stem cells (FOP-iPSCs) to elucidate the mechanisms underlying flare-ups and found that bone morphogenetic protein (BMP)-9 mediated enhanced proliferation by abnormal activation of transforming growth factor (TGF)-β signaling pathway in MSCs from FOP-iPSCs. In FOP model mice, elevated BMP-9 levels correlated with elevated phosphorylation of SMAD2/3 and increased cellular proliferation in the affected tissues, while systemic BMP-9 neutralization and knockout mitigated flare-ups and HO. Thus, BMP-9 aberrantly transduces TGF-β signaling and induces fibroproliferation, initiating flare-ups. This study provides novel insights into the development of future FOP therapies.

**Keywords** Fibrodysplasia Ossificans Progressive; Flare-Up; Fibroproliferation; Patient-Derived Induced Pluripotent Stem Cells; TGF-β Signaling
**Subject Categories** Genetics, Gene Therapy & Genetic Disease; Musculoskeletal System

See also: Li & Yuan

## Introduction

Fibrodysplasia ossificans progressiva (FOP, Mendelian Inheritance in Man [MIM] #135100) is a rare autosomal-dominant disorder characterized by incapacitating heterotopic ossification (HO) within skeletal muscle and connective tissues. This condition is often accompanied by recurrent and painful episodes of soft tissue swelling, which are referred to as flare-ups. Flare-ups are often triggered by trauma, including minor injuries such as a minor fall or injection, which can eventually lead to the development of new abnormal bone growth in the affected areas. The flare-up tissues include a substantial population of proliferative-mesenchymal stromal cells (MSCs) which potentially contribute to the ectopic bone formation through cartilage differentiation and endochondral ossification (Kaplan et al, 2005; Lees-Shepard et al, 2018b; Pignolo et al, 2016; Shore and Kaplan, 2010). Fibroadipogenic progenitors (FAPs), a subtype of MSCs residing in the interstitium of skeletal muscles, have recently been identified as the major cell-of-origin for HO in FOP mouse models (Lees-Shepard et al, 2018b; Zhao and Ikeya, 2024) and represent the specific proliferative cells in flare-up tissue.

FOP is driven by mutations in ACVR1, a bone morphogenetic protein (BMP) type I receptor. Approximately >95% of patients with FOP share a common R206H mutation within the intracellular glycine- and serine-rich domains of ACVR1 (Shore et al, 2006). This mutation leads to aberrant activation of the BMP signaling pathway, culminating in dysregulated MSCs/FAPs that initiate HO (Lees-Shepard et al, 2018b; Shore et al, 2006). A definitive treatment for FOP is currently elusive (Kitoh, 2020). Surgical interventions to remove the heterotopic bone frequently exacerbate the development of HO. Early-stage symptom management involves the use of anti-inflammatory drugs such as corticosteroids, and non-steroidal anti-inflammatory drugs only offer partial relief (Brantus and Meunier, 1998; Kitoh, 2020; Rhen and Cidlowski, 2005). Recent therapeutic advancements for FOP encompass diverse therapeutic agents, including the RARγ agonist palovarotene (Chakkalakal et al, 2016; Lees-Shepard et al, 2018a), direct kinase inhibitors targeting BMP type I receptors' catalytic domains (Engers et al, 2013; Hamasaki et al, 2012; Hao et al, 2010; Mohedas et al, 2013; Sanvitale et al, 2013; Yu et al, 2008), and neutralizing antibodies and inhibitors against Activin-A signaling and mTOR signaling (del Re et al, 2004; Hatsell et al, 2015; Hino et al, 2017; Hino et al, 2015; Sako et al, 2010; Souza et al, 2008). Although clinical trials are underway for candidates such as palovarotene (Pignolo et al, 2023), REGN2477 (Wang et al, 2023), rapamycin (Kaplan et al, 2018b), and saracatinib (Smilde et al, 2022), and recently palovarotene has been approved in Canada and the United States, the current landscape of targeted therapeutics has primarily

[1]Laboratory of Skeletal Development and Regeneration, Key Laboratory of Clinical Laboratory Diagnostics (Ministry of Education), College of Laboratory Medicine, Chongqing Medical University, Chongqing 400016, China. [2]Department of Clinical Application, Center for iPS Cell Research and Application (CiRA), Kyoto University, 53 Kawahara-cho, Shogoin, Sakyo-ku, Kyoto 606-8507, Japan. [3]Takeda-CiRA Joint Program, Fujisawa, Kanagawa, Japan. [4]Regenerative & Cellular Medicine Kobe Center, Sumitomo Pharma Co., Ltd., Konohana-ku, Osaka 554-0022, Japan. ✉E-mail: chengzhu.zhao@cqmu.edu.cn; mikeya@cira.kyoto-u.ac.jp

focused on addressing ectopic osteogenesis in the mid-to-late stages of the disease. Flare-ups inflict significant pain and distress on patients and are characteristic early symptoms of FOP. Moreover, the presence of retained MSCs within flare-up tissues increases the risk of HO recurrence (Stanley et al, 2022). Therefore, elucidation of the mechanisms underlying flare-ups is important for the development of FOP treatments.

Histological assessment reveals that the pathological process of flare-ups involves three major events: muscle degeneration, inflammatory cell infiltration, and subsequent fibroproliferative tissue formation, driven by a substantial population of MSCs/FAPs (Chakkalakal et al, 2012; Lees-Shepard et al, 2018a; Lees-Shepard et al, 2018b; Shore and Kaplan, 2010). Given the limitations of inflammatory inhibition in completely halting HO progression (Convente et al, 2018; Kaplan et al, 2018a; Pignolo et al, 2016; Wang et al, 2021; Zhang et al, 2020), our study aims to elucidate the mechanisms of MSCs/FAPs proliferation to better understand flare-up pathogenesis and offer insights for early FOP intervention. We examined the cell-proliferation rate as a flare-up indicator, utilizing MSCs obtained from FOP patient-derived induced pluripotent stem cells (FOP-iPSCs), which have been established in previous studies (Fukuta et al, 2014; Hino et al, 2017; Hino et al, 2015; Hino et al, 2018; Matsumoto et al, 2013; Matsumoto et al, 2015). The results obtained from the above model and FOP model mice harboring human *FOP-ACVR1 (R206H)* alleles demonstrated that BMP-9 aberrantly activates transforming growth factor (TGF)-β signaling and induces fibroproliferation, thereby initiating flare-ups. Furthermore, systemic BMP-9 neutralization and knockout mitigated both flare-ups and HO. These findings enhance our understanding and provide a new target for addressing the early stages of FOP.

# Results

## BMP-9 triggers aberrant proliferation in FOP-induced MSCs and FOP model mice with HO

To identify ligands that selectively stimulate MSC proliferation via FOP-ACVR1 but not WT-ACVR1, we employed induced MSCs derived from FOP-iPSCs (FOP-iMSCs) as experimental cells (Fukuta et al, 2014; Matsumoto et al, 2013). Mutation-rescued FOP-iMSCs (resFOP-iMSCs) were employed as genetically matched controls, which we developed and used in previous studies (Hino et al, 2017; Hino et al, 2018; Matsumoto et al, 2015). A series of TGF-β superfamily members were examined (Fig. 1A,B; Appendix Fig. S1). Ligands that stimulate TGF-β signaling, such as Activin-A, TGF-β1, and TGF-β3, significantly activated proliferation in both resFOP-iMSCs and FOP-iMSCs. On the other hand, ligands that stimulate BMP signaling, such as BMP-2, BMP-4, and BMP-7, exhibited negligible effects on resFOP/FOP-iMSC proliferation (Appendix Fig. S1). However, BMP-9 specifically activated the proliferation of FOP-iMSCs (Fig. 1B). This pro-proliferative effect showed a dose-dependent trend (Appendix Fig. S1), and this trend was consistently observed in the FOP-iMSCs derived from another patient (Appendix Fig. S2A). Similarly, BMP-9 treatment increased the proportion of Ki67-positive cells (Fig. 1C,D; Appendix Fig. S2B,C) and upregulated the cell cycle distribution within the G0/G1 phase, specifically in FOP-iMSCs (Fig. 1E; Appendix Fig. S3).

Subsequently, we assessed the in vivo effects of BMP-9 treatment. We had previously generated conditional transgenic mice harboring human *FOP-ACVR1 (R206H)* alleles (Hino et al, 2017). The transgenic mice conditionally expressed *hFOP-ACVR1 (R206H)* upon doxycycline (Dox) administration. Given that swelling is the most common presenting symptom of flare-ups (93%)(Pignolo et al, 2016) and considering the study's focus on fibroproliferative tissue formation during this phase, we have identified the appearance of swelling as the macroscopic indicator and the presence of proliferating FAPs/MSCs (Ki67+ PDGFRα-positive cells) as the histological indicator of flare-ups, as PDGFRα is widely recognized as the most specific single marker for identifying FAPs in muscle tissue (Uezumi et al, 2010; Agarwal et al, 2016; Wosczyna et al, 2019). Following intramuscular injection of BMP-9 into the gastrocnemius muscle (Fig. 1F), the induction of tissue resembling a flare-up (day 7) and HO (day 14) was observed in Dox-treated (Dox (+)) FOP mice, whereas no discernible HO formation occurred in Dox (−) FOP mice with BMP-9 injection, wild-type mice receiving Dox and BMP-9, and Dox (+) FOP mice with PBS injection (Fig. 1G; Appendix Fig. S4A–C). Specifically, the injection site of Dox (+) mice exhibited positive safranin O staining (reflecting the presence of acidic proteoglycan, an extracellular matrix protein of chondrocytes) on day 7 (Fig. 1G) and von Kossa staining (indicative of calcium deposition) on day 14 (Appendix Fig. S4D–F) post-BMP-9 injection. Immunohistochemical analysis revealed an abundance of Ki67-positive cells at the injection site in Dox (+) mice on day 7 post-injection, while minimal or undetectable staining was observed in Dox (−) mice (Fig. 1G,H). Collectively, these findings indicate that BMP-9 drives the aberrant proliferation of FOP cells in vitro and in vivo.

## Elevated BMP-9 levels during HO pathogenesis in CTX-induced FOP model mice

BMP-9 is abundantly expressed at sites of wound healing and tissue regeneration (Breitkopf-Heinlein et al, 2017). Clinical evidence has also indicated elevated BMP-9 expression around HO tissue in response to severe trauma (Grenier et al, 2013). To investigate the correlation between BMP-9 production and HO development in FOP tissues, FOP mice were administered a cardiotoxin (CTX) injection to induce muscle injury. This approach simulated trauma-induced flare-ups (Fig. 2A) (Chakkalakal et al, 2012). The observations revealed a slight elevation in circulating BMP-9 levels from day 1 after CTX treatment, which significantly increased by days 7 and 14 (Fig. 2B). Immunohistochemical observation and quantitative analysis at various stages of FOP lesion formation in Dox (+) mice (Fig. 2C,D) revealed that BMP-9-positive cells accumulated in the perivascular space before CTX injection (day 0). Following CTX-induced muscle destruction (day 1), these cells dispersed deep within the skeletal muscle and subsequently into early FOP lesions (days 3, 5, and 7), as well as ectopic cartilage and bone (day 14). The proliferating Ki67+ cells initially surrounded, then partly (days 1 and 3), and finally precisely (days 5 and 7) colocalized with BMP-9-positive cells. By day 14, cells at the injected site were embedded in a BMP-9-rich extracellular matrix, which may have contributed to the elevated serum BMP-9 levels observed at this stage. In the tissues of CTX-injected Dox (−) mice, the population of BMP-9-positive cells initially increased

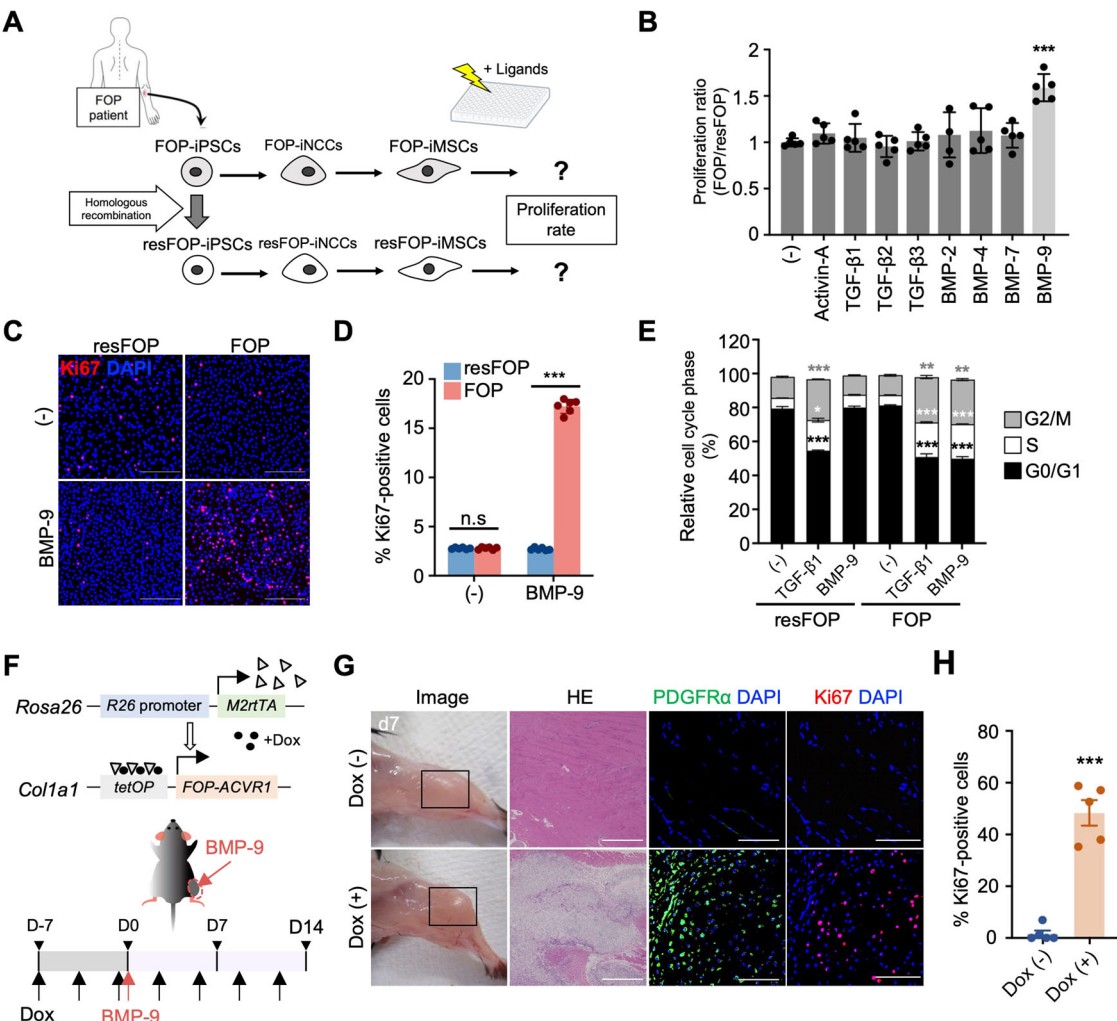

**Figure 1. Abnormal proliferation of FOP-iMSCs and flare-ups triggered by BMP-9 in FOP-ACVR1 transgenic mice.**

(A) Schematic representation of FOP-iMSC-specific ligand screening. (B) Aberrant activation of proliferation (FOP/resFOP) among the tested TGF-β superfamily ligands in the CCK-8 assay. Data represent the mean ± SD ($n = 5$ independent experiments). ***$P < 0.001$ ($P < 0.0001$) by one-way ANOVA with Dunnett's multiple comparisons test compared to the no ligand treatment control (−). (C) Representative images of Ki67 immunofluorescent staining. Scale bar, 100 μm. (D) Percentage of Ki67-positive cells relative to the total number of DAPI-stained nuclei. Data represent the mean ± SD ($n = 6$). Six randomly selected panels were analyzed, with at least 3950 cells counted per panel. n.s. no significant difference; ***$P < 0.001$ ($P < 0.0001$) by multiple t-tests in comparison with resFOP-iMSCs under the same condition. (E) Quantitative analysis of the proportions of cell cycle phases. The results represent the mean ± SD ($n = 3$ independent experiments). n.s., no significant difference; *$P < 0.05$; **$P < 0.01$; ***$P < 0.001$ (G0/G1: resFOP (−) vs TGF-β1 $P = 0.0006$, resFOP (−) vs FOP TGF-β1 $P = 0.0005$, resFOP (−) vs FOP BMP-9 $P < 0.0001$; S: resFOP (−) vs TGF-β1 $P = 0.0105$, resFOP (−) vs FOP TGF-β1 $P < 0.0001$, resFOP (−) vs FOP BMP-9 $P < 0.0001$; G2/M: resFOP (−) vs TGF-β1 $P < 0.0001$, resFOP (−) vs FOP TGF-β1 (−) $P = 0.0019$, resFOP (−) vs FOP BMP-9 $P = 0.0049$) by two-way ANOVA with Tukey's multiple comparisons test compared to the no ligand treatment control. (F) Schematic representation of the in vivo study using FOP-ACVR1 conditional transgenic mice. (G) Macroscopic observation and histological analysis of the BMP-9-injected region (day 7). Representative images of H&E staining (scale bar, 500 μm), and immunohistochemical staining for PDGFRα (green) and Ki67 (red) (scale bar, 100 μm) is shown. (H) Percentage of Ki67-positive cells relative to the number of nuclei. The results represent the mean ± SD ($n = 5$). Five randomly selected panels were analyzed, with an average of 150 cells counted per panel. ***$P < 0.001$ ($P < 0.0001$) by Student's t-test in comparison with the Dox (−) group. Source data are available online for this figure.

following muscle destruction (days 3 and 5) and then decreased as muscle regeneration proceeded (days 7 and 14) (Appendix Fig. S5A,B). Correspondingly, circulating BMP-9 levels in Dox (−) FOP mice did not significantly increase by days 7 and 14 (Appendix Fig. S5C). These findings suggest that the fundamental nature of FOP may lie in the breakdown of negative feedback mechanisms.

The immunohistochemical results of BMP-9 show its tissue distribution at days 3, 7, and 14 post-CTX injection (Fig. 2E). In

addition, the source of BMP-9 secretion was verified via co-staining with F4/80 (macrophage marker), PDGFRα (FAP marker), and SP7 (osteochondral lineages marker) (Hojo and Ohba, 2022) (Fig. 2F–H). Specifically, BMP-9 was found to colocalize with F4/80+ macrophages by day 3 post-injury (Fig. 2E,F), aligning with the peak recruitment period for immunocytes (Shore and Kaplan, 2010), with minimal co-expression in FAPs at this stage (Appendix Fig. S5D). In the context of early FOP lesions, the majority of PDGFRα+ FAPs begin to colocalize with BMP-9 (day 7, Fig. 2E,G),

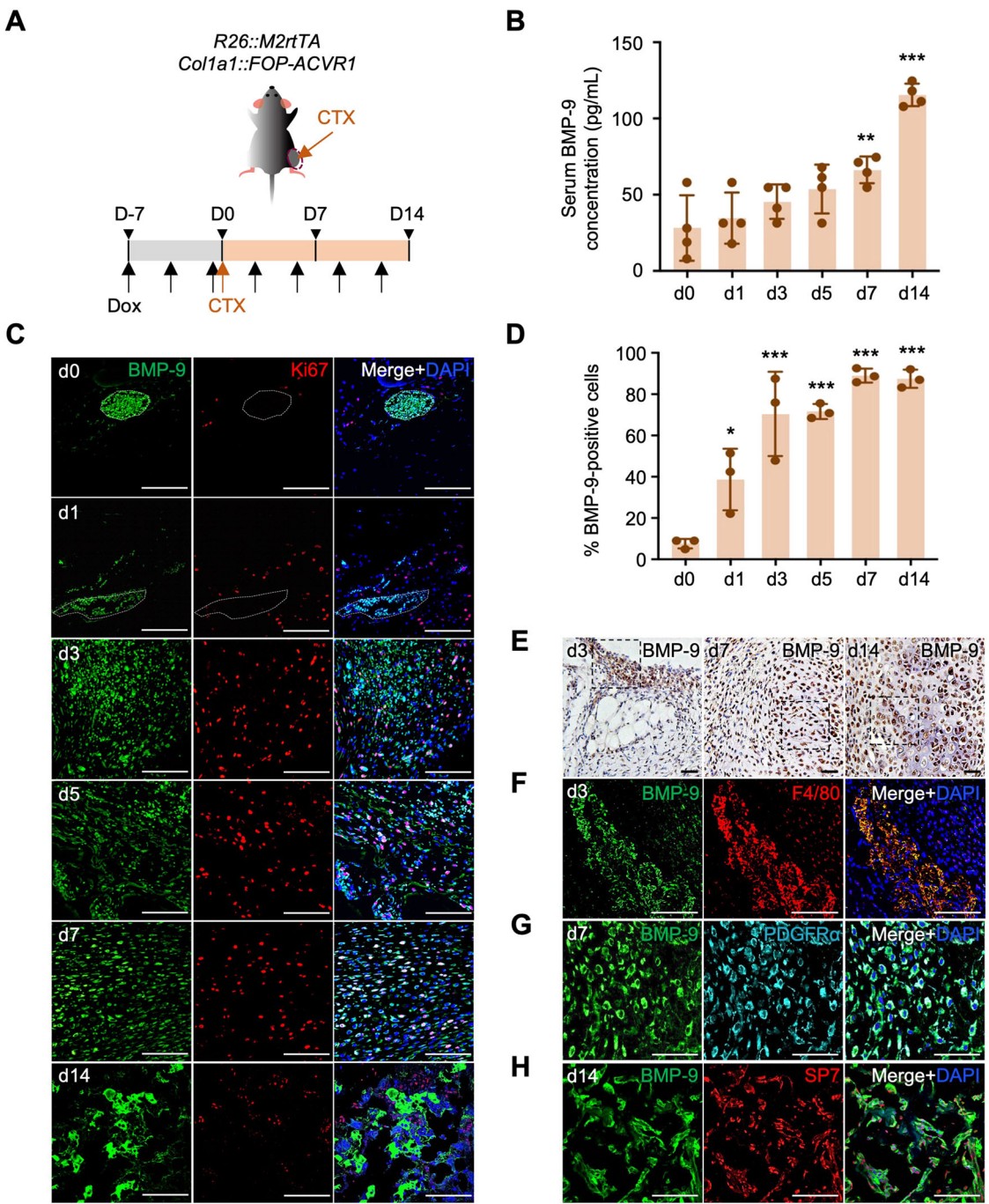

**Figure 2. BMP-9 expression and distribution in CTX-injected tissue of FOP mice.**

(A) Schematic diagram of the in vivo study. CTX injection induced HO. (B) Circulating BMP-9 levels in CTX-injected Dox (+) FOP mice. The results represent the mean ± SD ($n = 4$). **$P < 0.01$, ***$P < 0.001$ (d0 vs d7 $P = 0.0067$, d0 vs d14 $P < 0.0001$) by one-way ANOVA with Dunnett's multiple comparisons test compared to the CTX-untreated group (d0). (C) Immunohistochemical staining of BMP-9 (green) and Ki67 (red) at various time points (d1, 3, 5, 7, and 14). The BMP-9-positive cell clusters outlined by white dashed lines were observed under all conditions. Higher magnification in panel (E) shows these cells are F4/80-positive. Due to the excessive density of cells in these areas and their sparse and uneven distribution within the tissue, they were excluded from the quantification in panel (D). Scale bar, 100 μm. (D) Percentage of BMP-9-positive cells relative to the total number of DAPI-stained nuclei. The results represent the mean ± SD ($n = 3$). Three randomly selected panels were analyzed, with an average of 220 cells counted per panel. *$P < 0.05$, ***$P < 0.001$ (d0 vs d1 $P = 0.016$, d0 vs d3 $P < 0.0001$, d0 vs d5 $P < 0.0001$, d0 vs d7 $P < 0.0001$, d0 vs d14 $P < 0.0001$) by one-way ANOVA with Dunnett's multiple comparisons test compared to d0. (E) Immunohistochemical staining for BMP-9 in tissue at days 3, 7, and 14 post-CTX injection. (F–H) Co-immunofluorescence staining of BMP-9 (green) with (F) F4/80+ cells (red), (G) PDGFRα+ cells (light blue), and (H) SP7+ cells (red). Scale bar, 50 μm. Source data are available online for this figure.

while at the HO site, BMP-9 colocalized with the of the extracellular matrix surrounding SP7$^+$ osteochondral lineage cells (day 14, Fig. 2E,H). Collectively, these findings suggest that monocytes/macrophages expressing BMP-9 during the early stages of inflammation may contribute to FAP proliferation and BMP-9 production, leading to cell-autonomous expression of BMP-9 in FAPs and aberrant formation of HO in FOP model mice.

## *Bmp9-KO; hFOP-ACVR1* transgenic mice exhibit attenuated HO under CTX treatment

Since BMP-9 expression is correlated with the progression of lesions in FOP model mice, we next evaluated the role of endogenous BMP-9 in HO. Using CRISPR/Cas9-mediated genome editing, we generated *Bmp9-KO; hFOP-ACVR1* mice using fertilized oocytes carrying the compound heterozygous *ROSA26::M2rtTA* and *Col1a1::FOP-ACVR1* alleles (Fig. 3A; Appendix Fig. S6A–C). A previous report described that *Bmp9-KO* mice showed no obvious age-dependent phenotypes in comparison with their wild-type littermates (Bouvard et al, 2022). Similarly, our investigation revealed the viability of adult *Bmp9-KO; hFOP-ACVR1* (*Bmp9* KO) mice, and the normal appearance of their offspring (Appendix Fig. S6D,E). After administering Dox and injecting CTX to induce HO, a significant reduction in HO bone volume was observed at the CTX-injection site of *Bmp9-KO; hFOP-ACVR1* mice in comparison with WT; *hFOP-ACVR1* (WT) mice at 2 weeks (Fig. 3B,C). In contrast, the bone mineral density was comparable between the two groups (Appendix Fig. S6F), suggesting that BMP-9 had no impact on the quality of the ectopic bone tissue formed. At the CTX-injected site, BMP-9-positive cells were absent in *Bmp9-KO; hFOP-ACVR1* mice (Fig. 5D). Safranin O- and von Kossa-positive areas and COL2- and COL1-positive areas were present in both *Bmp9-KO; hFOP-ACVR1* and WT; *hFOP-ACVR1* mice; however, the extent of the positive areas were diminished in the former, corroborating the decrease in HO bone volume in *Bmp9-KO; hFOP-ACVR1* mice (Fig. 3E). These findings underscore the effectiveness of *Bmp9-KO* in suppressing HO in FOP mouse models even when a complete blockade of HO progression was not achieved.

## BMP-9 neutralizing antibody attenuates HO progression in CTX-injected FOP model mice

Subsequently, we investigated the potential effects of BMP-9 inhibition in an FOP mouse model to mitigate flare-ups and HO. BMP-9 neutralizing antibody (BMP-9 Ab) or isotype control antibody was subcutaneously administered to FOP model mice twice a week from the day of CTX injection for 1 week (0–1 W), 2 weeks (0–2 W), or 1 week after CTX injection for 1 week (1–2 W) (Fig. 4A). Notably, the administration of BMP-9 Ab led to the alleviation of early FOP lesion formation and the accumulation of fibroproliferative cells by day 7 post-CTX injection (Fig. 4B,C). BMP-9 Ab treatment significantly decreased the Ki67-positive PDGFRα$^+$ cell population, supporting its inhibitory effect on the proliferation of HO precursors. Moreover, by day 14 post-CTX injection, the administration of BMP-9 Ab in both the 0–1 W and 0–2 W groups substantially mitigated HO volume in comparison with that in mice treated with the control antibody. However, the administration of BMP-9 Ab 1 week post-CTX injection (1–2 W)

did not lead to a significant reduction in HO volume (Fig. 4D,E; Appendix Fig. S7A), indicating a window of opportunity (0–1 W) for effective HO suppression before the emergence of early FOP lesions. BMP-9 Ab did not alter the bone mineral density (Fig. S7C), suggesting that BMP-9 primarily promotes the proliferation of MSCs (chondroprogenitors), leading to increased bone volume at later stages, rather than directly affecting chondrogenesis and osteogenesis.

Histological analysis revealed positive staining for safranin O, von Kossa, COL2, and COL1 at the CTX injection site in mice treated with the control antibody. Administration of BMP-9 Ab within 0–1 W and 0–2 W after CTX injection resulted in a limited scope of ectopic cartilage and bone, effectively suppressing HO development (Fig. 4F). In contrast, mice treated with BMP-9 Ab between 1 and 2 W showed similar staining patterns to those treated with the control antibody. In these experiments, BMP-9 Ab administration did not lead to a reduction in body weight (Appendix Fig. S7B), implying that the abatement of HO may not be predominantly linked to toxicity. Taken together, the findings indicated that inhibition of BMP-9 during the early stage effectively mitigated initial FOP lesion formation and HO progression in FOP model mice.

## BMP-9 abnormally transduces TGF-β signaling in FOP-iMSCs and FOP model mice

To reveal the molecular mechanisms underlying BMP-9-induced proliferation, global gene expression analyses were performed using resFOP/FOP-iMSCs derived from FOP patients. Consistent with the in vitro and in vivo observation (Fig. 1), the top 20 gene ontology (GO) pathways showed specific activation of cell cycle-related pathways in FOP-iMSCs in comparison with resFOP-iMSCs after BMP-9 treatment (Fig. 5A). Notably, PCA plots revealed that BMP-9 treatment showed a substantial shift toward TGF-β-like signaling in FOP-iMSCs in comparison with resFOP-iMSCs, while treatment with other ligands did not (Fig. 5B). Gene set perturbation analysis using Kyoto Encyclopedia of Genes and Genomes (KEGG) data of the regulated pathway indicated that BMP-9 treatment exerts distinct effects on cell cycle regulation in FOP-iMSCs (Fig. 5C). TGF-beta pathway was also significantly upregulated with a $P$ value <0.001, although the change appears less pronounced compared to the other two pathways. Considering that the analysis results may be influenced by the algorithms used, heatmaps constructed with representative genes for the cell cycle, DNA replication, and the TGF-β signaling pathway were presented, further supporting the upregulation of these pathways in FOP-iMSCs (Fig. 5D).

To further investigate whether BMP-9 induces abnormal TGF-β signaling in FOP-iMSCs, we conducted experiments using a TGF-β-sensitive luciferase reporter construct (CAGA-Luc) (Hino et al, 2015). This construct was transfected into both FOP-iMSCs and resFOP-iMSCs, and luminescence was measured 16 h after ligand stimulation. Remarkably, BMP-9 treatment elicited a dose-dependent increase in luciferase activity in FOP-iMSCs, but not in resFOP-iMSCs (Fig. 5E,F). FOP-iMSCs specifically showed heightened expression of downstream genes in the TGF-β signaling pathway (Fig. 5G), as well as phosphorylation of SMAD2/3, which are cytoplasmic transducers of TGF-β signaling (Fig. 5H). Abundant phosphorylated SMAD2/3 positive cells were also

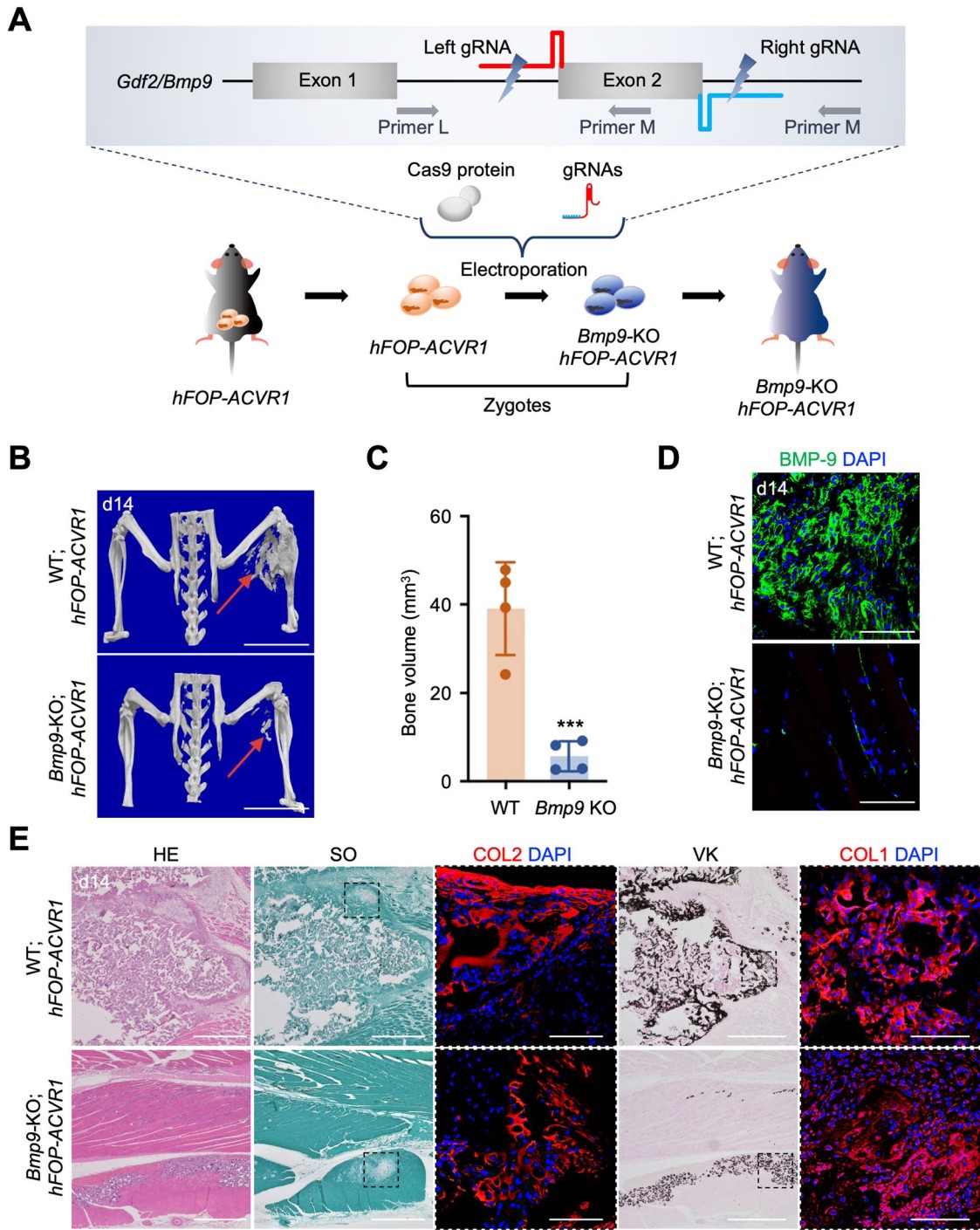

**Figure 3.** ***Bmp9*-KO-*hFOP-ACVR1* transgenic mice exhibit attenuated HO after CTX injection.**

(**A**) Schematic of the generation of *Bmp9*-KO-*hFOP-ACVR1* transgenic mice. CRISPR/Cas9-mediated *Bmp9*-KO of the target sequence from exon 2, which encoded the entire mature C-terminal region of BMP-9, is shown. (**B**) Representative μCT observations after CTX-induced HO at d14. Scale bar, 10 mm. (**C**) Heterotopic bone volume (mm$^3$) of each group. The results represent the mean ± SD ($n = 4$ mice). ***$P < 0.001$ ($P = 0.0009$) by Student's $t$-test in comparison with the WT-*hFOP-ACVR1* group. (**D**) BMP-9 cells were absent at the injected site in *Bmp9*-KO-*hFOP-ACVR1* mice. Scale bar, 50 μm. (**E**) Histological analysis of the CTX-injected region, including H&E, safranin O, von Kossa (scale bar, 500 μm), anti-COL2, and anti-COL1 staining (scale bar, 100 μm). Source data are available online for this figure.

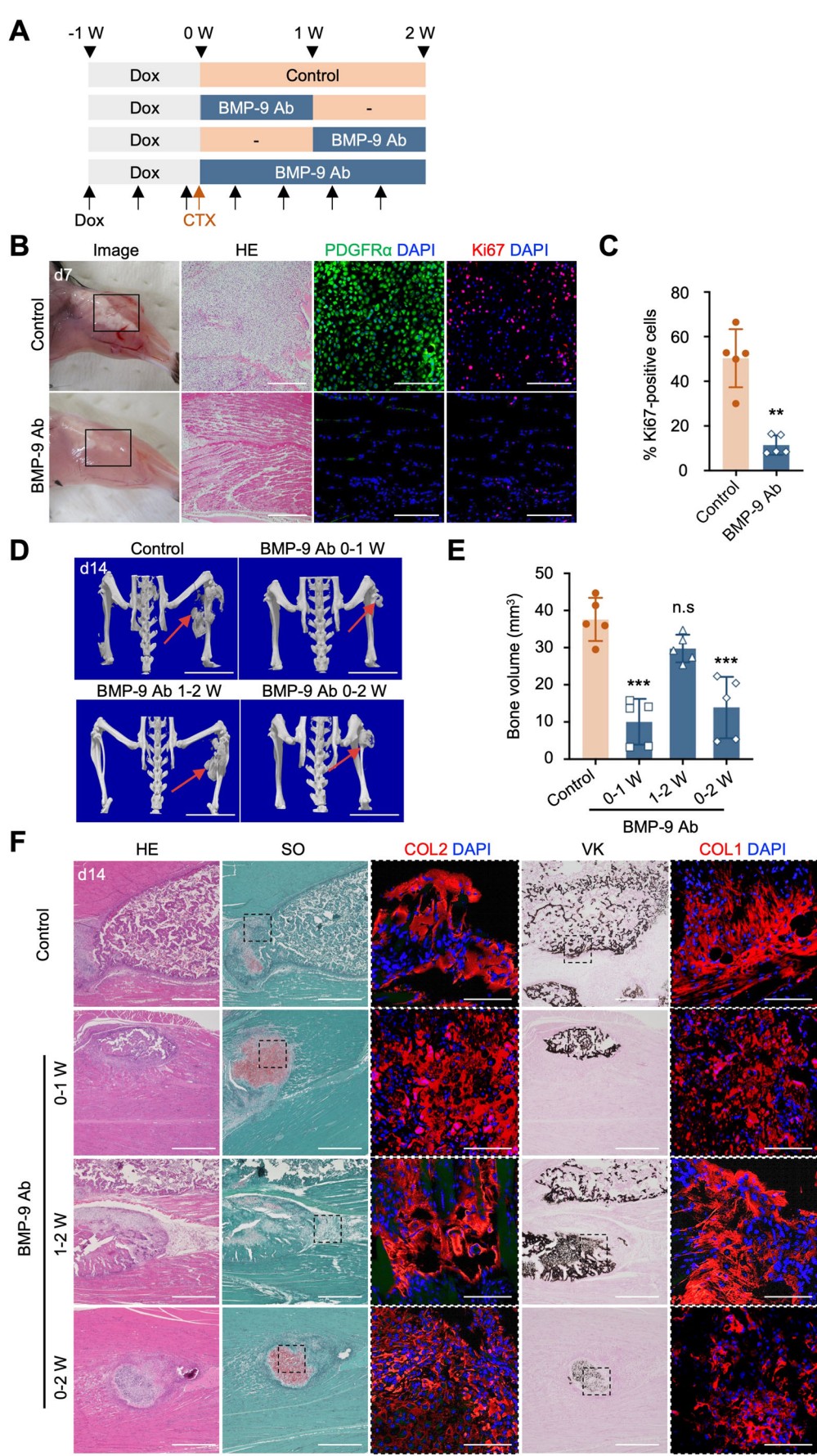

**Figure 4. Systemic BMP-9 neutralizing antibody administration mitigates early FOP lesions and HO progression in CTX-induced FOP mice.**

(A) Illustration depicting the strategy for administering Dox, CTX, and antibodies in FOP-ACVR1 conditional transgenic mice. Mice received subcutaneous injections of 10 mg/kg BMP-9 neutralizing antibody (BMP-9 Ab) or IgG2B isotype control antibody (Control) twice a week, starting from CTX injection for 1 week (0–1 W), 2 weeks (0–2 W), or 1 week after CTX injection for 1 week (1–2 W). (B) Representative images of macroscopic observation, H&E staining (scale bar, 500 μm), and immunohistochemical staining for PDGFRα (green) and Ki67 (red) (scale bar, 100 μm) in the CTX-injected region (day 7) are shown. (C) Percentage of Ki67-positive cells relative to the number of nuclei. The results represent the mean ± SD ($n = 5$). Five randomly selected panels were analyzed, with an average of 220 cells counted per panel. **$P < 0.01$ ($P = 0.0016$) by Student's *t*-test in comparison with the Dox (−) group. (D) Representative μCT observations at day 14 post-CTX injection. The red arrows indicate induced HO tissue. Scale bar, 10 mm. (E) Heterotopic bone volume (mm³) in each group. The results represent the mean ± SD ($n = 5$ mice). n.s., no significant difference; ***$P < 0.001$ (Control vs BMP-9 Ab 0–1 W $P < 0.0001$, Control vs BMP-9 Ab 0–2 W $P < 0.0001$) by one-way ANOVA with Dunnett's multiple comparisons test compared to the control antibody-treated group. (F) Histological analysis of the CTX-injected region, including H&E, safranin O, von Kossa (scale bar, 500 μm), anti-COL2, and anti-COL1 staining (scale bar, 100 μm). Source data are available online for this figure.

observed at the BMP-9 injected site in Dox (+) mice on day 7 and day 14 post-injection, whereas no detectable staining was observed in Dox (−) mice and *Bmp9-KO-hFOP-ACVR1* transgenic mice (Fig. 5I,J).

Since BMP-9 typically transduces BMP-SMAD1/5/9 signaling, we analyzed SMAD1/5/9 phosphorylation and the expression of downstream BMP signaling genes. Both levels were comparable in FOP-iMSCs and resFOP-iMSCs (Appendix Fig. S8). Moreover, an equivalent degree of activation of the BMP-sensitive luciferase reporter construct (BRE-Luc) in BMP-9-treated FOP- and resFOP-iMSCs has been confirmed in a previous study (Hino et al, 2015). These outcomes collectively affirm that BMP-9 specifically transduces TGF-β signaling in FOP-iMSCs.

Subsequent loss-of-function investigations through siRNA-mediated targeting of type I receptors validated the indispensability of FOP-ACVR1 (ALK2) for BMP-9-dependent TGF-β signaling (Fig. 5K). Similarly, siRNAs specific for type II receptors revealed the involvement of ACVR2A, BMPR2, and AMHR2 in this anomalous activation (Fig. 5L). These findings collectively indicate that BMP-9 induced aberrant TGF-β signaling through FOP-ACVR1 with ACVR2A and BMPR2.

**Inhibition of FOP-ACVR1-SMAD2/3 signaling suppresses BMP-9-induced proliferation in FOP-iMSCs**

BMP-9 induces abnormal activation of both the TGF-β signaling pathway and proliferation in FOP-iMSCs. To investigate the influence of TGF-β signaling on the heightened proliferation of FOP-iMSCs, we performed pharmacological inhibition and siRNA-mediated inhibition of this pathway. DMH-1, an inhibitor that disrupts the interaction within ACVR1's SMAD-binding pocket, was employed to identify ACVR1's contributions to heightened proliferation. Application of TGF-β1 resulted in comparable increases in cell-proliferation rates of both FOP- and resFOP-iMSCs (Fig. 6A). This augmentation was not influenced by DMH-1 treatment, implying that ACVR1 does not participate in TGF-β1-induced proliferation of iMSCs. Notably, DMH-1 treatment effectively attenuated the BMP-9-induced proliferation of FOP-iMSCs, indicating that FOP-ACVR1 specifically mediates the BMP-9-induced proliferation in iMSCs. In contrast, SB431542, an inhibitor that selectively blocks ALK4, 5, and 7 (also referred to as ACVRL1, TGFBR1, and ACVR1C) from binding to TGF-β ligands and inhibits downstream signaling, reduced TGF-β1-induced proliferation but had no significant effect on BMP-9-specific mediated proliferation of FOP-iMSCs. This result, together7 with Fig. 5K, indicates that ACVR1 likely mediates the

activation of TGF-β signaling and the proliferation of FOP-iMSCs directly, without involving other Type I receptors.

In addition to receptor inhibitors, a knockdown assay of cytoplasmic transducers of TGF-β signaling (SMAD2/3) and BMP signaling (SMAD1/5/9) was performed. siRNAs specifically targeting SMAD2/3 effectively curtailed the augmented proliferation induced by both TGF-β1 and BMP-9 (Fig. 6B,D; knockdown efficiencies are illustrated in Appendix Fig. S9), suggesting that TGF-β signaling is instrumental in mediating both TGF-β1 and BMP-9 induced proliferation. Moreover, attenuation of SMAD1/5/9 levels in relation to TGF-β1 and BMP-9-dependent proliferation activation was not evident (Fig. 6C). These findings collectively confirm the correlation between FOP-ACVR1-mediated TGF-β signaling and the aberrant proliferation of FOP-iMSCs. However, additional research is necessary to fully grasp the mechanisms underlying the abnormal activation of TGF-β signaling and proliferation by BMP-9.

## Discussion

Faced with the challenge of obtaining clinical samples from patients with FOP owing to surgical risks, our previous study developed a precise experimental system using MSCs produced from patient-derived iPSCs. This approach revealed that Activin-A enhances chondrogenesis in FOP-iMSCs via abnormal BMP signaling activation (BRE-Luc activity and phosphorylation of SMAD1), alongside the usual TGF-β pathway, contributing to the endochondral ossification in FOP (Hino et al, 2015). In this study, we found that BMP-9 abnormally increased proliferation in FOP-iMSCs by excessively activating TGF-β signaling (evidenced by increased CAGA-Luc activity and phosphorylation of SMAD2/3), in addition to its effects on the BMP signaling pathway. This result was consistent with the observations in FOP model mice, wherein elevated BMP-9 expression correlated with higher p-SMAD2/3 levels and increased cell proliferation.

We designed experiments to validate the effects of BMP-9 on FOP progression through targeted inhibition. Systemic administration of a BMP-9 neutralizing antibody following CTX-induced muscle injury notably decelerated the fibroproliferation and progression of HO in the FOP model mice. Antibody administration during the muscle destruction-fibroproliferation phase (0–1 week) has demonstrated significant therapeutic effects. However, it is ineffective when administered during the chondrogenesis or osteogenesis phases (1–2 weeks), likely due to redundancy from other BMPs. This suggests the highly stage-

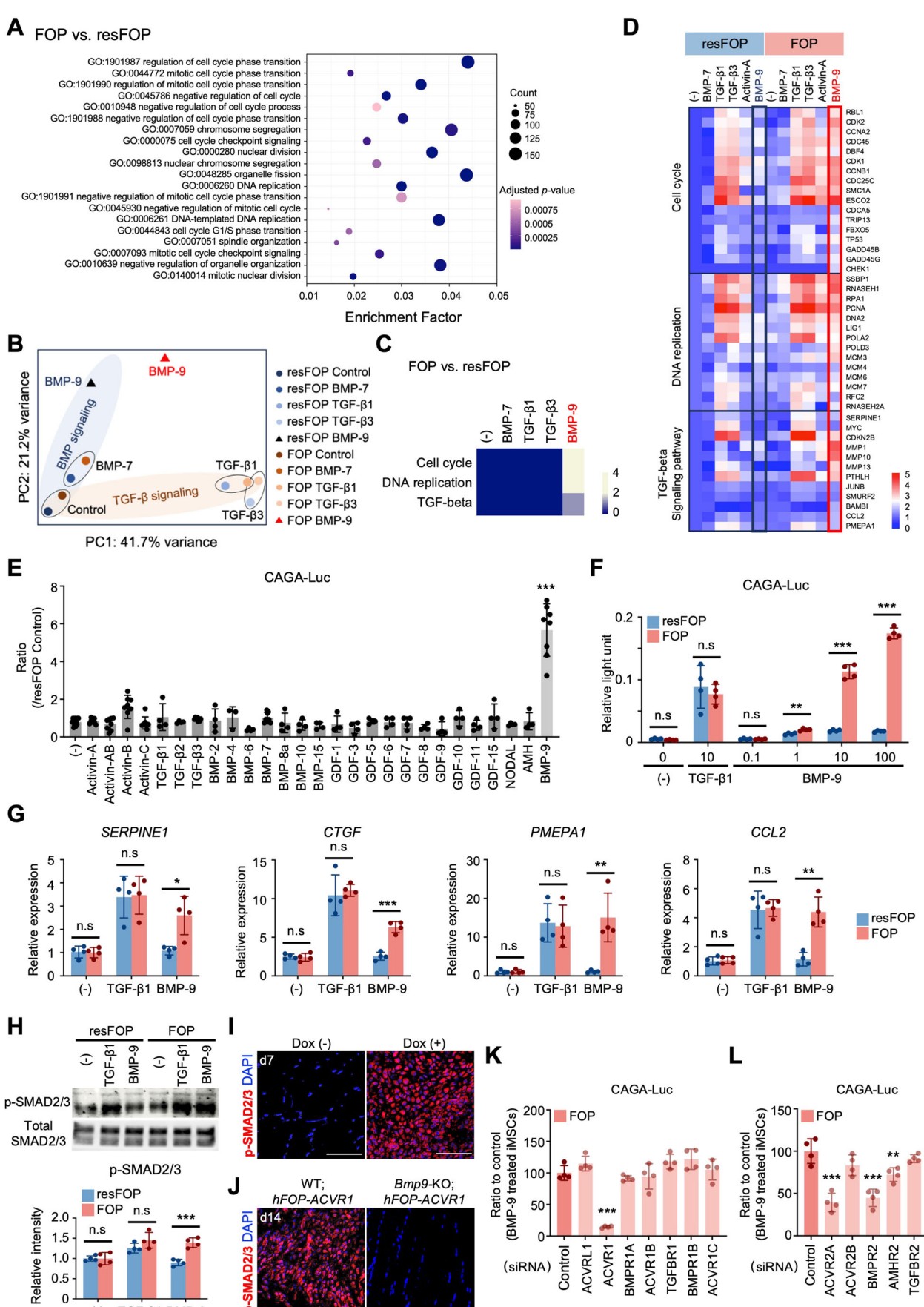

**Figure 5. BMP-9 abnormally transduced TGF-β signaling in FOP-iMSCs.**

(A–D) Global gene expression analysis showed that BMP-9 specifically activated cell cycle-related pathways and transduced TGF-β-like signaling in FOP-iMSCs. (A) The top 20 gene ontology (GO) pathways were significantly regulated using differentially expressed gene sets (FOP-iMSCs vs. resFOP-iMSCs, fold change >2), with significance determined by Fisher's exact test. (B) PCA plot of FOP- and resFOP-iMSCs. (C) Whole-gene-set perturbation analysis of the regulated pathways identified by Kyoto Encyclopedia of Genes and Genomes (KEGG) pathway analysis of the FOP-iMSCs in comparison with the resFOP-iMSCs following ligand treatment. Heatmap shows $-\log_{10}$-transformed $P$ values, calculated using Fisher's exact test for pathway enrichment significance. (D) Heatmap illustrating that BMP-9 specifically activated gene expression in the cell cycle, DNA replication, and the TGF-β signaling pathway in FOP-iMSCs. Data showing $\log_{10}$ (fold change) values related to no ligand treatment control (−) in resFOP-iMSCs. (E) BMP-9 induced the highest increase in CAGA-Luc activity (FOP/resFOP) among tested TGF-β superfamily ligands. All ligands were used at a concentration of 10 ng/mL. The results represent the mean ± SD ($n > 4$ independent experiments). ***$P < 0.001$ (Control vs BMP-9, $P < 0.0001$) by one-way ANOVA with Dunnett's multiple comparisons test compared to the no ligand treatment control. (F) BMP-9 increased CAGA-Luc activity in FOP-iMSCs, not in resFOP-iMSCs. The results represent the mean ± SD ($n = 4$ independent experiments). n.s., no significant difference; **$P < 0.01$; ***$P < 0.001$ (resFOP vs FOP: BMP-9 1 $P = 0.0025$, BMP-9 10 $P = 0.0004$, BMP-9 100 $P < 0.0001$) by multiple $t$-tests in comparison with resFOP-iMSCs under the same condition. (G) Elevated expression of TGF-β target genes in BMP-9-stimulated FOP-iMSCs in quantitative PCR (qPCR) analysis. ($n = 4$ independent experiments). n.s., no significant difference; *$P < 0.05$; **$P < 0.01$; ***$P < 0.001$ (resFOP vs FOP BMP-9: SERPINE1 $P = 0.0119$, CTGF $P = 0.0001$, PMEPA1 $P = 0.0044$, CCL2 $P = 0.0012$) by multiple $t$-tests in comparison with resFOP-iMSCs under the same condition. (H) Representative western blot image and quantification. After 6 h of serum starvation, FOP- and resFOP-iMSCs were treated with ligands for 1 h for subsequent analysis. BMP-9 induced phosphorylation of SMAD2/3 (p-SMAD2/3) in FOP-iMSCs, not in resFOP-iMSCs. The results represent the mean ± SD ($n = 4$ independent experiments). n.s., no significant difference; ***$P < 0.001$ (resFOP vs FOP BMP-9 $P = 0.0005$) by multiple $t$-tests in comparison with resFOP-iMSCs under the same condition. (I) Representative images of immunohistochemical staining for p-SMAD2/3 (red) from BMP-9-injected region at d7. Scale bar, 50 μm. (J) p-SMAD2/3 positive cells were absent at the CTX-injected site in Bmp9-KO-hFOP-ACVR1 mice at d14. Scale bar, 50 μm. (K, L) BMP-9 transduced TGF-β signaling through ALK2 (FOP-ACVR1), ACVR2A, and BMPR2. FOP-iMSCs transiently transfected with CAGA-Luc, CMV-Renilla, and siRNAs specific for type I receptors (K) or type II receptors (L) were stimulated with BMP-9 for 16 h. The results represent the mean ± SD ($n = 4$ independent experiments).). **$P < 0.01$; ***$P < 0.001$ (K: Control vs ACVR1 $P < 0.0001$; L: Control vs ACVR2A $P < 0.0001$, Control vs BMPR2 $P < 0.0001$, Control vs AMHR2 $P = 0.007$) by one-way ANOVA with Dunnett's multiple comparisons test compared to the control siRNA transfected-FOP-iMSCs. Source data are available online for this figure.

dependent nature of the treatment. These findings underscore the importance of early intervention during the critical fibroproliferation phase and highlight the necessity for stage-specific treatment strategies. In FOP patients, the recurrence of HO after surgery is primarily due to muscle damage. The results from the CTX-induced FOP model indicate that inhibiting intrinsic BMP-9 could potentially address the recurrence of HO. Further investigation is valuable to confirm the therapeutic potential of BMP-9 inhibition in preventing HO recurrence. Additionally, BMP-9 neutralizing antibodies alone were not sufficient to completely halt HO, and even genetic knockout of Bmp9 in FOP-ACVR mice reduced, rather than eliminated, HO progression. This finding emphasizes the need for combination therapy to achieve ideal treatment outcomes.

Recent therapeutic approaches for FOP, such as rapamycin targeting the mTOR signaling pathway, have been primarily effective during cartilage formation (Hino et al, 2017; Kaplan et al, 2018b). Treatments such as the RARγ agonist palovarotene (Pignolo et al, 2023), neutralizing antibodies like REGN2477 (anti-Activin-A mAb), and ACVR1 inhibitors (Wang et al, 2023; Davis et al, 2024) primarily suppress ACVR1 signaling, which plays a key role in inhibiting both cartilage and bone formation. These drugs act at various stages of the Activin-ACVR1-SMAD1 pathway, and their prolonged use can effectively inhibit HO. However, since each of the Activin-A, ACVR1, and SMAD1 pathways also regulates homeostasis, continuous and strong inhibition could potentially lead to side effects. Given these considerations, combination therapy might be worth exploring. For instance, long-term but minimal use of inhibitors targeting the Activin-ACVR1-SMAD1 pathway, with the addition of BMP-9 inhibitors at the onset of flare-up signs, and the use of rapamycin when cartilage formation is detected, could offer a promising avenue for effective FOP treatment.

Elevated TGF-β activity has been demonstrated in clinical specimens from patients with FOP and acquired HO (Barruet et al, 2018; Micha et al, 2016; Wang et al, 2018). Recent findings indicate that inflammatory conditions initiate HO formation by promoting TGF-β signaling activation in MSCs within soft tissues (Sorkin et al, 2020). Additionally, systemic administration of a TGF-β neutralizing antibody has been shown to inhibit HO in FOP mouse models (Wang et al, 2018). However, the exact relationship between TGF-β signal activation and HO in FOP remains unclear. Our results suggest that intrinsic BMP-9-induced TGF-β signal activation mediates FAP hyperproliferation, coinciding with the onset of flare-ups and HO in FOP model mice. These findings provide new insights into FOP pathogenesis.

However, the mechanism by which BMP-9 activates TGF-β signaling in FOP cells remains unclear. Our results suggest that the aberrant TGF-β signaling induced by BMP-9 is mediated through the FOP-ACVR1 type I receptor and requires the involvement of ACVR2A and BMPR2 type II receptors. Recent data from a novel ACVR1 inhibitor, BLU-782, indicate that blocking ACVR1 in the early stages of FOP alleviates the fibroproliferative response and reduces the bone volume of HO (Davis et al, 2024). This finding aligns with the results of BMP-9 neutralizing antibody administration, supporting the hypothesis that FOP-ACVR1 may mediate the activation of TGF-β signaling, thereby promoting the proliferation of MSCs/FAPs. Further research is necessary to fully understand the molecular mechanisms underlying the abnormal activation of TGF-β signaling through FOP-ACVR1 by BMP-9.

In summary, our study provides evidence for BMP-9-mediated abnormal proliferation in FOP-MSCs/FAPs through the activation of TGF-β signaling. Inhibition of BMP-9 during the early fibroproliferation stage was effective in mitigating the pathogenesis of flare-ups and HO in FOP. A deeper understanding of the process by which FOP-ACVR1 propagates TGF-β signaling could offer promising avenues for novel therapeutic interventions in FOP, and may also provide valuable insights for research on and treatment of acquired HO.

# Methods

### Reagents and tools table

| Reagent/resource | Reference or source | Identifier or catalog number |
|---|---|---|
| **Experimental models** | | |
| FOP patient-derived iPSCs | Yoshihisa Matsumoto | N/A |
| h*FOP-ACVR1* conditional transgenic mice | Yasuhiro Yamada | N/A |
| **Antibodies** | | |
| Ki67 (D3B5) Rabbit mAb | Cell Signaling | #12202 |
| BMP-9 (H-3) Mouse mAb | Santa Cruz Biotechnology | sc-514211 |
| Collagen II Ab-2 (Clone 2B1.5) Mouse mAb | Thermo Fisher Scientific | #MS-235-B0 |
| Collagen I alpha 1 Antibody | Novus Biologicals | NB600-408 |
| Anti-F4/80 antibody [CI: A3-1] | Abcam | ab6640 |
| Anti-PDGFR alpha antibody [RM0004-3G28] | Abcam | ab51875 |
| Anti-SP7/Osterix antibody | Abcam | ab22552 |
| Phospho-Smad2 (Ser465/467)/ Smad3 (Ser423/425) (D27F4) Rabbit mAb | Cell Signaling Technology | #8828 |
| Smad2/3 (D7G7) XP® Rabbit mAb | Cell Signaling Technology | #8685 |
| Phospho-Smad1 (Ser463/465)/ Smad5 (Ser463/465) / Smad9 (Ser465/467) (D5B10) Rabbit mAb | Cell Signaling Technology | #13820 |
| Smad1/5/8 Antibody (N-18)-R | Santa Cruz Biotechnology | sc-6031-R |
| Anti-rabbit IgG, HRP-linked Antibody | Cell Signaling Technology | #7074 |
| Goat anti-Mouse IgG (H + L) Secondary Antibody, Alexa Fluor® 488 conjugate | Invitrogen | A-28175 |
| Goat anti-Rabbit IgG (H + L) Cross-Adsorbed Secondary Antibody, Alexa Fluor® 555 conjugate | Invitrogen | A-21428 |
| Goat anti-Mouse IgG (H + L) Cross-Adsorbed Secondary Antibody, Alexa Fluor® 555 | Invitrogen | A-21422 |
| Goat anti-Rat IgG (H + L) Cross-Adsorbed Secondary Antibody, Alexa Fluor® 555 | Invitrogen | A-21434 |
| Goat anti-Rat IgG (H + L) Cross-Adsorbed Secondary Antibody, Alexa Fluor® 647 | Invitrogen | A-21247 |
| **Chemicals, enzymes, and other reagents** | | |
| Activin-A | R&D Systems | 338-AC |
| BMP-7 | R&D Systems | 354-BP |
| BMP-9 | R&D Systems | 3209-BP |
| TGF-β1 | R&D Systems | 240-B |
| SB431542 | Selleck Chemicals | 301836-41-9 |
| CHIR99021 | Axon Medchem | #1386 |
| DMH-1 | Tocris Bioscience | 4126 |
| Primate embryonic stem (ES) cell medium | ReproCELL | RCHEMD001 |
| FGF2 | Wako Pure Chemical | 064-04541 |
| Recombinant human EGF | R&D Systems | 236-EG |
| αMEM | Thermo Fisher Scientific | 12571063 |
| FBS | Nichirei | 174012 |
| Penicillin and streptomycin | Thermo Fisher Scientific | 15140122 |
| Cell counting kit 8 | Dojindo | CK04 |
| Paraformaldehyde | Nacalai | 09154-56 |
| Cell Cycle Assay Solution | Dojindo | C548 |
| Accutase | Innovative Cell Technologies | A1110501 |
| Liberate Antibody Binding Solution | COSMO BIO Co., Ltd. | 24310 |
| Blocking One | Nacalai Tesque | 03953-95 |
| Can Get Signal Immunostain Solution B | Toyobo | NKB-601 |
| Tween-20 | Sigma-Aldrich | P9416 |
| Cardiotoxin | Latoxan | L8102 |
| GDF2/BMP-9 ELISA Kit | LifeSpan BioSciences | LS-F3779 |
| pRL-CMV-Renilla luciferase reporter | Promega Corporation | E2261 |
| Dual-luciferase reporter assay system | Promega | E1910 |
| Silencer® Select Pre-designed siRNA | Thermo Fisher Scientific | |
| ECL Prime Western Blotting Detection Reagent | GE Healthcare | 12316992 |
| RNeasy Kit | QIAGEN | 74104 |
| DNAse-One Kit | QIAGEN | 79254 |
| Superscript III Reverse Transcriptase | Thermo Fisher Scientific | 18080093 |
| Thunderbird SYBR qPCR Mix | TOYOBO | QPS-201 |
| Human/Mouse/Primate BMP-9 monoclonal antibody | R&D Systems | MAB3209 |
| Mouse IgG2B isotype control antibody | R&D Systems | MAB004 |
| TrueGuide sgRNA | Invitrogen | A35534 |
| TrueCut Cas9 Protein v2 | Invitrogen | A36497 |
| Opti-MEM media | Gibco | 31985062 |
| DNeasy Blood & Tissue kit | QIAGEN | 69504 |
| KOD-neo-plus polymerase | Toyobo, Co., Ltd. | KOD-401 |
| **Software** | | |
| FlowJo | BD Biosciences | |
| TRI/3D-BON software | Ratoc System Engineering | |
| Image Lab™ | Bio-Rad Laboratories | |
| GeneSpring GX | Agilent Technologies | |
| Ingenuity Pathway Analysis | QIAGEN | |
| **Other** | | |
| Aria II instrument | BD Biosciences | |

| Reagent/resource | Reference or source | Identifier or catalog number |
|---|---|---|
| μFX-1000 | Fujifilm | |
| DX-50 | Faxitron Bioptics | |
| InspeXio SMX-100CT | Shimadzu | |
| BZ-X710 | KEYENCE | |
| Zeiss LSM 710 | Zeiss | |
| EnVision® Multilabel Reader | PerkinElmer Co., Ltd | |
| Bio-Rad Molecular Imager Chemi-Doc™ XRS+ | Bio-Rad Laboratories | |
| StepOne Real-Time PCR System | Applied Biosystems | |
| Ion S5 XL System | Thermo Fisher Scientific | |

## Cell culture

The FOP patient-derived iPSCs used in this study (from patients 1 and 2, previously described as vFOP4-1 and vFOP5-22 (Matsumoto et al, 2013), respectively) harbored the *R206H* heterozygous mutation in *ACVR1*, and gene-corrected resFOP-iPSCs were generated by BAC-based homologous recombination (Matsumoto et al, 2015). All the experiments shown in Figs. 1, 5, 6; Appendix Figs. S1, 3, 8, 9 were performed using FOP-iPSCs from patient 1 and resFOP-iPSCs (cl1). The data for the other clones are shown in Appendix Fig. S2. These cells fulfilled several criteria for iPSCs, including the expression of pluripotent markers, teratoma formation, and normal karyotype and morphology. The growth and gene expression profiles of the resFOP-iPSC clones were indistinguishable from those of the original FOP-iPSCs (Matsumoto et al, 2015).

iPSCs were maintained in primate embryonic stem (ES) cell medium (ReproCELL) supplemented with 4 ng/mL recombinant human FGF2 (Wako Pure Chemical). The induction and maintenance of induced neural crest cells (iNCCs) and iMSCs derived from iPSCs have been previously described (Fukuta et al, 2014; Matsumoto et al, 2015). Briefly, to activate the production of iNCCs, mTeSR1 medium (STEMCELL Technologies) was used for feeder-free iPSC culturing. iNCCs were induced in chemically defined medium (CDM) supplemented with 10 μM SB431542 and 1 μM CHIR99021 for 7 days and then maintained in CDM supplemented with 10 μM SB431542, 20 ng/mL FGF2, and 20 ng/mL recombinant human EGF (R&D Systems). The iNCCs were used for up to 20 passages. iMSCs were induced and maintained in αMEM (Invitrogen, Thermo Fisher Scientific) supplemented with 10% (v/v) fetal bovine serum (FBS; Nichirei), 5 ng/mL FGF2, and 0.5% penicillin and streptomycin (Invitrogen, Thermo Fisher Scientific). The iMSCs in passages 3–6 were used in this study.

## TGF-β superfamily ligands and related reagents

All TGF-β superfamily ligands were purchased from R&D Systems Inc. (Minneapolis, KA, USA), except growth differentiation factor (GDF)-6, GDF-7 (GeneTex Inc., Irvine, CA, USA), and GDF-15 (Abnova Corporation, Taipei, Taiwan). All the ligands were

dissolved according to the manufacturer's instructions. DMH-1 was purchased from Tocris Bioscience (Bristol, UK). BMP-9 and BMP-7 were used at concentrations of 100 ng/mL, and TGF-β1 was used at a concentration of 10 ng/mL unless otherwise noted.

## Cell-proliferation assay

For the quantitative assessment of proliferation, FOP- and resFOP-iMSCs were seeded at 2500 cells/well in 96-well plates and left to adhere overnight. The cells were then treated with TGF-β superfamily ligands. After 72 h of incubation, the cell proliferation rates were measured using the cell counting kit 8 (CCK-8) assay. For Ki67 immunofluorescence staining, cells were seeded at 5000 cells/well in 24-well plates and treated with BMP-9 for 72 h. The plates were fixed in 4% paraformaldehyde (Nacalai, Japan) at room temperature (around 25 °C) for 10 min, washed with phosphate-buffered saline (PBS), and permeabilized with 0.5% Triton X-100 in PBS for 10 min. The blocking and antibody staining steps were the same as those described in the section on Histological analysis.

## Cell cycle assay

A cell cycle assay solution (C548, Dojindo) was used to measure the cell cycle according to the manufacturer's instructions. Briefly, after 72 h of incubation with ligands, cells were dissociated with Accutase (Innovative Cell Technologies, San Diego, CA, USA) and incubated with a cell cycle assay solution for 15 min at 37 °C. The DNA content was determined by measuring the staining intensity using an Aria II instrument (BD Biosciences, Franklin Lakes, NJ, USA). Cell cycle distribution was analyzed using the FlowJo software (BD Biosciences) according to the manufacturer's protocol.

## BMP-9-injection using *hFOP-ACVR1* conditional transgenic mice

h*FOP-ACVR1* conditional transgenic mice were generated according to a previously reported method (Beard et al, 2006; Ohnishi et al, 2014; Yamada et al, 2013). Briefly, *FOP-ACVR1*, followed by *IRES-mCherry*, was targeted into the 3′-UTR region of the *Col1a1* gene locus under the tetracycline-dependent promoter of KH2 ES cells, which harbors the optimized reverse tetracycline-dependent transactivator at the *ROSA26* locus (Ohnishi et al, 2014; Yamada et al, 2013). After confirming germline transmission, The offspring resulting from the cross between homozygous *ROSA26::M2rtTA* mice and homozygous *tetO-FOP-ACVR1* mice, carrying both alleles, were used to induce the h*FOP-ACVR1* gene.

Female mice (age- and body weight-matched between groups) between 16 and 20 weeks of age were used unless otherwise noted. Mice were administered 2 mg/mL doxycycline (Dox) in drinking water supplemented with 10 mg/mL sucrose to induce h*FOP-ACVR1*. To confirm the role of BMP-9 in FOP tissue, BMP-9 (1 μg/mouse) was injected into the right gastrocnemius muscle of Dox-treated or untreated mice. The mice were euthanized and analyzed on days 7 and 14 post-injection.

For obtaining X-ray images, mice were anesthetized with isoflurane (5% for induction, 2–3% for maintenance; Abbvie), which was followed by immobilization and X-ray imaging using μFX-1000 (Fujifilm) or DX-50 (Faxitron Bioptics). Micro-CT

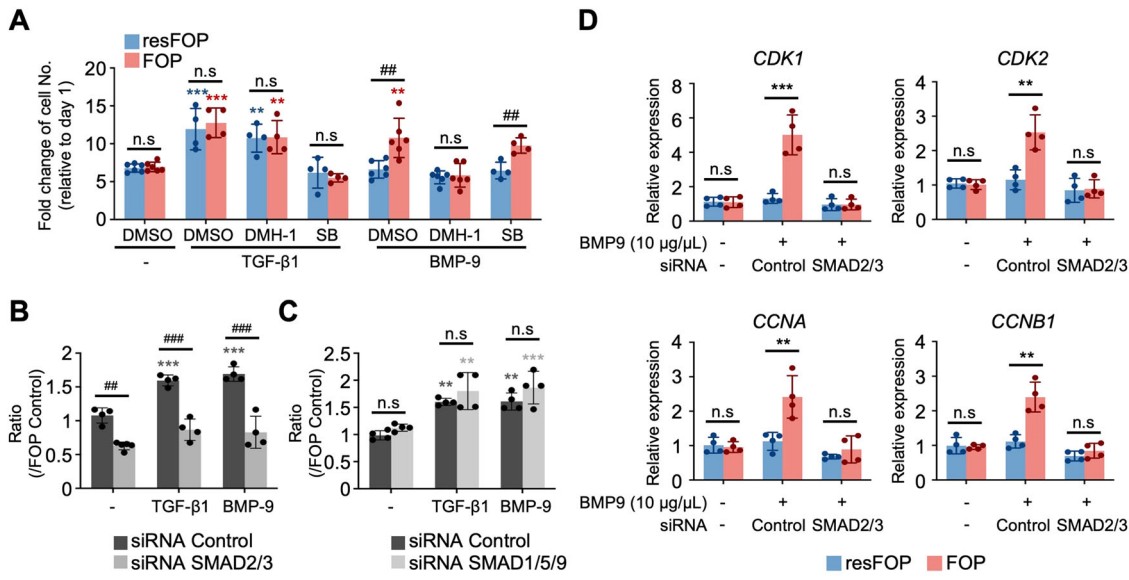

**Figure 6. Inhibition of TGF-β signaling suppresses BMP-9-induced proliferation in FOP-iMSCs.**

(A) Proliferation of BMP-9-triggered FOP-iMSCs was suppressed by 1 μM DMH-1, but not by 25 μM SB431542 (SB) treatment. Proliferation rates of stimulated cells were evaluated via the CCK-8 assay. The results represent the mean ± SD (n = 4–6 independent experiments). n.s., no significant difference; **P < 0.01; ***P < 0.001 (resFOP: (−) vs TGF-β1 DMSO P < 0.0001, (−) vs TGF-β1 DMH-1 P = 0.0023, FOP: (−) vs TGF-β1 DMSO P < 0.0001, (−) vs TGF-β1 DMH-1 P = 0.0067, (−) vs BMP-9 DMSO P = 0.0029) compared to the no treatment control; ##P < 0.01 (resFOP vs FOP BMP-9: DMSO P = 0.009, SB P = 0.0047) compared to resFOP-iMSCs under the same condition, analyzed using two-way ANOVA with Tukey's multiple comparisons. (B, C) FOP-iMSCs transiently transfected with siRNAs specific for SMAD2/3 (B) or SMAD1/5/9 (C) were stimulated with BMP-9. The results represent the mean ± SD (n = 4 independent experiments). ***P < 0.001 (B: siRNA Control: (−) vs TGF-β1 P < 0.0001, (−) vs BMP-9 P = 0.0001; C: siRNA Control: (−) vs TGF-β1 P = 0.0065, (−) vs BMP-9 P = 0.0059; siRNA SMAD1/5/8: (−) vs TGF-β1 P = 0.0027, (−) vs BMP-9 P = 0.001) compared to the no ligand treatment control; n.s., no significant difference; ##P < 0.01; ###P < 0.001 (B: siRNA Control vs SMAD2/3: (−) P = 0.0031, TGF-β1 P < 0.0001, BMP-9 P < 0.0001) compared to resFOP-iMSCs under the same condition, analyzed using two-way ANOVA with Tukey's multiple comparisons. (D) Expression profiles of genes related to cellular proliferation in BMP-9-stimulated resFOP- and FOP-iMSCs in qPCR analysis. The results represent the mean ± SD (n = 4 independent experiments). n.s., no significant difference; **P < 0.01, ***P < 0.001 (resFOP vs FOP BMP-9 siRNA Control: CDK1 P = 0.0008, CDK2 P = 0.0032, CCNA P = 0.0054, CCNB1 P = 0.0085) by multiple t-tests in comparison with resFOP-iMSCs under the same condition. Source data are available online for this figure.

images were obtained using an X-ray CT system (InspeXio SMX-100CT; Shimadzu). Three-dimensional reconstruction and determination of the volume of heterotopic bone were performed using TRI/3D-BON software (Ratoc System Engineering) according to the manufacturer's instructions.

## Histological analysis

At the indicated time points, tissue samples were collected, fixed overnight in 4% paraformaldehyde, and embedded in paraffin. The blocks were sectioned at 5-μm intervals. Hematoxylin and eosin (H&E), safranin O, and von Kossa staining were performed as described previously (Hino et al, 2017; Hino et al, 2018). For immunohistochemistry, the deparaffinized sections were rehydrated and pretreated with Liberate Antibody Binding Solution (COSMO BIO Co., Ltd., Tokyo, Japan) for antigen retrieval. After washing with PBS, the tissue sections were blocked with Blocking One (Nacalai Tesque) for 60 min and then incubated with primary antibodies diluted in Can Get Signal Immunostain Solution B (Toyobo) for 16 to 18 h at 4 °C. Next, the samples were washed with 0.2% Tween-20 (Sigma-Aldrich) in PBS and incubated with Alexa Fluor 488-, 555-, or 647-conjugated secondary antibodies (Thermo Fisher Scientific) diluted in Can Get Signal Immunostain Solution B for 1 h at room temperature. DAPI (10 μg/ml) was used to counterstain nuclei. The samples were observed using a BZ-X710

(KEYENCE) or Zeiss LSM 710 (Zeiss) confocal microscope. The primary and secondary antibodies used in this study are summarized in Appendix Table S1.

## Establishment of cardiotoxin-induced HO model in *hFOP-ACVR1* conditional transgenic mice

For the FOP mouse model, cardiotoxin (CTX, 9.1 μg/mouse, Latoxan) was injected into the right gastrocnemius muscle to initiate skeletal muscle injury and subsequent heterotopic bone formation (Chakkalakal et al, 2012). The mice were euthanized and analyzed on days 0, 1, 3, 5, 7, and 14 post-injection.

## Serum BMP-9 analysis

The concentration of BMP-9 in mouse serum was determined using a GDF2/BMP-9 ELISA Kit (LifeSpan BioSciences, LS-F3779) in accordance with the manufacturer's instructions.

## Luciferase assay

The CAGA-Luc reporter was constructed as described previously (Dennler et al, 1998). The BRE-Luc reporter was purchased from Addgene (Cambridge, MA) (Korchynskyi and ten Dijke, 2002). pRL-CMV-Renilla luciferase reporter (Promega Corporation,

Madison, WI, USA) was transiently transfected and used for normalization. Luciferase activity was measured using a dual-luciferase reporter assay system (Promega), as described previously (Hino et al, 2015). The luminescence signal was measured on EnVision® Multilabel Reader (PerkinElmer Co., Ltd, Waltham, MA, USA) according to the manufacturer's instructions. Small interference RNA (siRNAs) specific for type I or II receptors and the negative control were purchased from Thermo Fisher Scientific Inc. (Silencer® Select Pre-designed siRNA, Waltham, MA, USA) as described previously (Hino et al, 2015). Detailed information is provided in Appendix Table S2.

## Western blotting

SDS-PAGE and blotting with whole-cell lysates were performed using standard procedures. Protein bands were detected using ECL Prime Western Blotting Detection Reagent (GE Healthcare, Little Chalfont, UK) and visualized using a Bio-Rad Molecular Imager Chemi-Doc™ XRS+ with Image Lab™ software (Bio-Rad Laboratories, Inc., Hercules, CA, USA). The antibodies used in this study are listed in Appendix Table S1. All data (relative intensity) were corrected using total SMADs.

## Quantitative polymerase chain reaction analysis

Total RNA was purified using the RNeasy Kit (QIAGEN) and treated with the DNAse-One Kit (QIAGEN) to remove genomic DNA. Total RNA (0.3 µg) was reverse-transcribed for single-stranded cDNA using random primers and Superscript III Reverse Transcriptase (Thermo Fisher Scientific) according to the manufacturer's instructions. Quantitative polymerase chain reaction (qPCR) was performed using the Thunderbird SYBR qPCR Mix (TOYOBO) and analyzed using the StepOne Real-Time PCR System (Applied Biosystems). The primer sequences used are listed in Appendix Table S3.

## Global gene expression analyses

FOP- and resFOP-iMSCs were stimulated with 100 ng/mL of BMP-7, BMP-9, or Activin-A, or with 10 ng/mL of TGF-β1 or TGF-β3. After incubation for 72 h, mRNA was extracted. Total RNA was purified using an RNeasy Micro Kit (QIAGEN) and treated with a DNAse-One Kit (QIAGEN) to remove genomic DNA. We reverse-transcribed 10 ng of total RNA to obtain single-stranded cDNA using a SuperScript VILO cDNA Synthesis Kit (Thermo Fisher Scientific). We performed cDNA library synthesis for the Ion Ampliseq transcriptome using the Ion AmpliSeq Transcriptome Human Gene Expression Core Panel (Thermo Fisher Scientific) and Ion Ampliseq Library Kit Plus (Thermo Fisher Scientific) according to the manufacturer's protocol. Briefly, cDNA was amplified over 12 cycles with Ion AmpliSeq™ Transcriptome Human Gene Expression Core Panel by a thermal cycler. Primer sequences were partially digested with FuPa reagent sequentially for 10 min at 50 °C, 10 min at 55 °C, and 20 min at 60 °C. Barcode-labeled cDNA libraries were analyzed using the Ion S5 XL System (Thermo Fisher Scientific) and Ion 540 Chip Kit (Thermo Fisher Scientific). PCA and hierarchical clustering were performed using the GeneSpring GX software. Pathway and upstream analyses were performed using Ingenuity Pathway Analysis (QIAGEN).

## Neutralizing antibody treatment

For the neutralizing antibody treatment experiments, mice were subcutaneously injected with Human/Mouse/Primate BMP-9 monoclonal antibody (MAB3209; R&D Systems) or mouse IgG2B isotype control antibody (MAB004; R&D Systems) at 10 mg/kg body weight, twice a week from the day of CTX injection for 1 week (0–1 W), 2 weeks (0–2 W), or 1 week after CTX injection for 1 week (1–2 W). The mice were euthanized and analyzed at 1 and 2 weeks after CTX treatment.

## Generation of *Bmp9-KO-hFOP-ACVR1* transgenic mice

We generated *Bmp9*-KO-h*FOP-ACVR1* transgenic mice using CRISPR/Cas9-mediated genome editing. Fertilized oocytes with a heterozygous *ROSA26::M2rtTA* allele and a tetO-h*FOP-ACVR1* allele to induce the *FOP-ACVR1* gene were generated via in vitro fertilization (IVF) (Nakagawa et al, 2016). The fertilized oocytes were used in subsequent experiments.

The single-guide RNA (sgRNA) design was optimized to maximize cutting specificity and efficiency using the online KOnezumi freeware (Kuno et al, 2019). CRISPR knockout was performed by removing exon 2, which encodes the entire mature C-terminal region of BMP-9. The sequences we selected (Appendix Fig. S6A) were unique to the genome, avoiding the presence of predicted off-target sites. Customized RNA was synthesized by Invitrogen (TrueGuide sgRNA; A35534). Ribonucleoprotein complexes (RNP) consisting of 0.2 µg/µL Cas9 protein (TrueCut Cas9 Protein v2) and 0.1 µg/µL gRNAs were reconstituted in 50 µL of Opti-MEM media (Gibco). For electroporation, the zygotes were aligned in a chamber with platinum plate electrodes (CUY5001P1-1.5, NEPA GENE Co. Ltd) filled with 5 µL of RNP-containing medium. After multi-pulse (20–25 V, 3-ms length, 100-ms intervals) electroporation (NEPA GENE Co. Ltd., Chiba, Japan), the zygotes were removed, washed in prewarmed M2 media, and surgically transferred into pseudopregnant ICR female mice.

To determine the genotype of the offspring, genomic DNA was extracted from the tail lysates of pups using a commercial kit (DNeasy Blood and Tissue kit, QIAGEN), and PCR was performed using KOD-neo-plus polymerase (Toyobo, Co., Ltd.) with each primer set (Appendix Fig. S6A). Next, 18-week-old *Bmp9* KO and littermate WT control mice were administered Dox and cardiotoxin, euthanized, and analyzed 2 weeks after treatment.

## Statistical analysis

At least three independent experiments were performed. Comparisons between the two groups were conducted using the unpaired *t*-test. For comparisons involving more than two groups, one-way ANOVA was employed, followed by Dunnett's multiple comparison test when compared to the control group. For comparisons between two groups under multiple conditions, multiple *t*-tests were utilized. For comparisons involving two factors and three or more groups, two-way ANOVA followed by Tukey's multiple comparison test was employed. Differences between groups were considered statistically significant at a *P* value of <0.05. Statistical analyses were conducted using GraphPad Prism 8.0 (GraphPad Software Inc., San Diego, CA, USA).

**The paper explained**

**Problem**

Fibrodysplasia ossificans progressiva (FOP) is a rare genetic disorder presenting with progressive heterotopic ossification (HO) in soft tissues. While clinical trials are underway for therapeutic candidates, current targeted therapies have mainly focused on addressing ectopic osteogenesis in the mid-to-late stages of the disease. Early-stage FOP is characterized by recurrent episodes of painful tissue swelling (flare-ups), and the mechanisms underlying these flare-ups remain currently unclear. The flare-up tissues contain a substantial population of proliferative-mesenchymal stromal cells (MSCs) that potentially contribute to ectopic bone formation and increase the risk of HO recurrence. Therefore, elucidation of the mechanisms underlying flare-ups is important for the development of FOP treatments.

**Results**

We demonstrate that bone morphogenetic protein (BMP)-9 mediated the enhanced proliferation of MSCs obtained from FOP patient-derived induced pluripotent stem cells (FOP-iPSCs), through abnormal activation of the transforming growth factor (TGF)-β signaling pathway. In FOP model mice, elevated BMP-9 levels correlated with elevated phosphorylation of SMAD2/3 and MSC proliferation in flare-up tissues, while systemic BMP-9 neutralization and knockout mitigated flare-ups and HO.

**Impact**

Our findings highlight the significant impact of BMP-9-induced TGF-β signaling on MSC over-proliferation in FOP flare-ups. Interventions targeting BMP-9 or TGF-β signaling elements may present a therapeutic option for early-stage FOP, offering valuable insights into advancing FOP treatment strategies.

## Study approval

All experimental protocols involving human participants were conducted in accordance with the principles outlined in the World Medical Association (WMA) Declaration of Helsinki and the Department of Health and Human Services Belmont Report. Written informed consent was obtained from all participants prior to their involvement in the study, and all procedures were approved by the Ethics Committee of the Department of Medicine and Graduate School of Medicine at Kyoto University. All the animal experiments were approved by the Institutional Animal Committee of Kyoto University, Kyoto, Japan.

## Data availability

The datasets generated during this study are available in the supporting information. RNA-seq data that support the findings have been deposited in the NCBI Sequence Read Archive (SRA) database under accession number PRJNA1166719; available at https://www.ncbi.nlm.nih.gov/sra/PRJNA1166719, with access beginning 2025-10-02. The imaging dataset has been deposited in BioImage Archive under accession number S-BIAD1412; available at https://doi.org/10.6019/S-BIAD1412, with access beginning 2025-10-09.

The source data of this paper are collected in the following database record: biostudies:S-SCDT-10_1038-S44321-024-00174-3.

## Peer review information

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

## Acknowledgements

We thank the MI laboratory members for their support and stimulating discussions. We thank A. Tanaka from the Animal Research Facility of CiRA for technical assistance in generating *Bmp9*-KO-*hFOP-ACVR1* transgenic mice. We thank the CiRA Common Equipment Management Office for providing the research instruments such as FACS and qPCR. This work was supported by the iPS Cell Research Fund, and Japan Agency for Medical Research and Development (AMED) under Grant Number JP15bm0104001 and 23bm1323001 to MI. This work was also supported by grants-in-aid for scientific research from the Japan Society for the Promotion of Science (JSPS) (Nos. 19K16540 and 21K06855), the Young Scientists Fund of the National Natural Science Foundation of China (Grant No. 82202655), and Natural Science Foundation of Chongqing (CSTB2024NCSQ-MSX0457) to CZ.

## Author contributions

**Chengzhu Zhao**: Conceptualization; Data curation; Formal analysis; Funding acquisition; Writing—original draft; Project administration; Writing—review and editing. **Yoshiko Inada**: Data curation; Formal analysis. **Souta Motoike**: Data curation; Formal analysis. **Daisuke Kamiya**: Data curation; Formal analysis. **Kyosuke Hino**: Data curation; Formal analysis; Supervision; Writing—review and editing. **Makoto Ikeya**: Conceptualization; Supervision; Funding acquisition; Writing—original draft; Writing—review and editing.

Source data underlying figure panels in this paper may have individual authorship assigned. Where available, figure panel/source data authorship is listed in the following database record: biostudies:S-SCDT-10_1038-S44321-024-00174-3.

## Disclosure and competing interests statement

K. Hino is an employee of Sumitomo Pharma Co. Ltd. The remaining authors declare no competing interests.

