## [Peer Review File · EMBO Molecular Medicine]

BMP-9 mediates fibroproliferation in fibrodysplasia ossificans progressiva through TGF- β signaling

Chengzhu Zhao, Yoshiko Inada, Souta Motoike, Daisuke Kamiya, Kyosuke Hino, and Makoto Ikeya

Corresponding author(s): Makoto Ikeya (mikeya@cira.kyoto-u.ac.jp) , Chengzhu Zhao (chengzhu.zhao@cqmu.edu.cn)

Review Timeline:

Revision Received:	10th Jun 24
Editorial Decision:	1st Jul 24
Revision Received:	9th Oct 24
Editorial Decision:	23rd Oct 24
Revision Received:	29th Oct 24
Accepted:	31st Oct 24

Editor: Zeljko Durdevic

Transaction Report:

1st Jul 2024

Dear Prof. Ikeya,

Thank you for the submission of your manuscript to EMBO Molecular Medicine. We have now received feedback from the three reviewers who agreed to evaluate your manuscript. All three referees recognize interest of the study but also raise important concerns that should be addressed in a major revision. If you would like to discuss further the points raised by the referees, I am available to do so via email or video. Let me know if you are interested in this option.

We would welcome the submission of a revised version within three months for further consideration. Please let us know if you require longer to complete the revision.

I look forward to receiving your revised manuscript.

Yours sincerely,

Zeljko Durdevic

We require:

- 1) A .docx formatted version of the manuscript text (including legends for main figures, EV figures and tables). Please make sure that the changes are highlighted to be clearly visible.
- 2) Individual production quality figure files as .eps, .tif, .jpg (one file per figure). For guidance, download the 'Figure Guide PDF': (<https://www.embopress.org/page/journal/17574684/authorguide#figureformat>).
- 3) A .docx formatted letter INCLUDING the reviewers' reports and your detailed point-by-point responses to their comments. As part of the EMBO Press transparent editorial process, the point-by-point response is part of the Review Process File (RPF), which will be published alongside your paper.
- 4) A complete author checklist, which you can download from our author guidelines (<https://www.embopress.org/page/journal/17574684/authorguide#submissionofrevisions>). Please insert information in the checklist that is also reflected in the manuscript. The completed author checklist will also be part of the RPF.
- 5) Please note that all corresponding authors are required to supply an ORCID ID for their name upon submission of a revised manuscript.
- 6) It is mandatory to include a 'Data Availability' section after the Materials and Methods. Before submitting your revision, primary datasets produced in this study need to be deposited in an appropriate public database, and the accession numbers and

database listed under 'Data Availability'. Please remember to provide a reviewer password if the datasets are not yet public (see <https://www.embopress.org/page/journal/17574684/authorguide#dataavailability>).

13) Author contributions: You will be asked to provide CRediT (Contributor Role Taxonomy) terms in the submission system. These replace a narrative author contribution section in the manuscript.

14) A Conflict of Interest statement should be provided in the main text.

15) Every published paper now includes a 'Synopsis' to further enhance discoverability. Synopses are displayed on the journal

webpage and are freely accessible to all readers. They include a short stand first (maximum of 300 characters, including space) as well as 2-5 one-sentences bullet points that summarizes the paper. Please write the bullet points to summarize the key NEW findings. They should be designed to be complementary to the abstract - i.e. not repeat the same text. We encourage inclusion of key acronyms and quantitative information (maximum of 30 words / bullet point). Please use the passive voice. Please attach these in a separate file or send them by email, we will incorporate them accordingly.

16) Include a Reagents and Tools Table as part of the Methods section, which can be downloaded from our author guidelines (<https://www.embopress.org/page/journal/17574684/authorguide#structuredmethods>)

**** Reviewer's comments ****

Referee #1 (Comments on Novelty/Model System for Author):

This study employs patient-derived iPSCs and FOP model mice to investigate the role of BMP-9 in mediating fibroproliferation in FOP. The authors used Rosa26::M2rtTA;Col1A1:FOP-ACVR1 mice as the FOP model

Referee #1 (Remarks for Author):

In this manuscript, Zhao et al. demonstrates the involvement of intrinsic BMP-9 in the fibroproliferation of BOP through the abnormal activation of TGF- β signaling. This study employs patient-derived iPSCs and FOP model mice to investigate the role of BMP-9 in mediating fibroproliferation in FOP. Genetic knockout of BMP-9 and administration of BMP-9 neutralizing antibodies significantly mitigate heterotopic ossification, highlighting BMP-9 as a promising therapeutic target for FOP. This provides new insights into potential therapeutic strategies for mitigating HO in FOP. I have a few comments that the authors may take into consideration to improve their manuscript.

Major concerns:

1. The authors suggested that BMP-9 primarily regulated the fibroproliferation of BOP. However, it seems that both chondrogenic and osteogenic capacities were also affected in Bmp9-KO mice (Figure 3E) and in mice treated with the BMP-9 neutralizing antibody (Figure 4E). In addition, the intensity of Safranin O staining was increased in the BMP9 Ab (0-2w) treatment mice. However, the expression of COL2 and COL1 in the BMP9 Ab (1-2w) treatment mice seemed to be minimally affected compared to the control. I think these data need to be quantified, and the authors should examine and explain the impact on chondrogenic and osteogenic capacities.
2. HO cannot be treated through surgical procedures due to its recurrence at the surgical site. Can Bmp9-KO or the application of Bmp-9 Ab alleviate the recurrence of HO after resection?
3. In Line 146, the authors stated, "monocytes/macrophages expressing BMP-9 during the early stages of inflammation may contribute to FAP proliferation and BMP-9 production, leading to autocrine expression of BMP9 in FAPs." How do the authors confirm the autocrine expression of BMP9 in FAPs?

Major concern :

1. The abbreviation "FOP" should not be used in the title.
2. The full term should be written out at the first time an abbreviation appears in the main text, even if it has already been defined in the abstract.
3. In line 155, the authors stated, "Similarly, our investigation revealed the viability of adult Bmp9-KO; hFOP-ACVR1 (Bmp9 KO) mice and the normal appearance of their offspring (SI Appendix, Fig. S6 D and E)." However, in Figure 3B, it appears that the femur length of BMP9-KO mice is shortened.
4. In Figure 2G, the trends of increased expression of SERPINE1, PMEPA1, and CCL2 in the BMP9 treatment group are inconsistent with the trends shown in the heatmap in Figure 2D.
5. DAPI immunofluorescence needs to provide, e.g. Figures 1G, 3D, 5I, and 5J.
6. The gene and protein names are not consistently formatted.
 - "Col1A1" should be written as "Col1a1" (Figures 1F, 2A).
 - "p-Smad2/3" in Figures 5I and 5J should be "p-SMAD2/3".

Referee #2 (Comments on Novelty/Model System for Author):

The combination of animal models (BMP9-KO and Cre-induced ACVR1) and the patient-derived iPS cells is sufficient.

Referee #2 (Remarks for Author):

Zhao et al. investigated the molecular mechanisms leading to FOP in soft tissue. Focusing on the early stages of FOP, the authors screened which BMP induces FOP and identified BMP9 as the triggering factor that leads to MSC proliferation and tissue swelling. Using BMP9 KO cells, the authors further demonstrated that endogenous BMP9 is involved in the progression of FOP. The injection of a neutralising antibody against BMP9 supports the idea that blocking BMP9 is effective in inhibiting bursting FOP. RNA sequencing and subsequent experimental analysis (immunohistochemistry and reported assays) further confirmed that BMP9 is a major extracellular molecule that induces MSC proliferation. Finally, the authors showed that BMP9 signal is critically mediated by Smad2/3 to induce the aberrant expression of the TGF β -targeted genes.

Effectively using the combination of animal models and patient-derived iPS cells, the overall study is well conducted and the analyses were thoroughly done. The authors provide a number of compelling data and thus convincingly demonstrate the usefulness of BMP9 in the treatment of FOP. Especially Fig3 and 4 are well-presented and provide hope to patients suffering from this disease.

However, there are several points that I would like the authors to clarify and discuss before making a final decision on publication in the journal.

- 1) The purpose of the study is to describe how the "painful tissue swelling (flare-ups)" occurs - however, there is no description or data in the results section. If the Fig1G images show the "flare-up" phenotype, the authors must clearly describe and explain this to ensure that the animal model used in this study properly recapitulates the phenotypes of FOP.
- 2) The authors' group has previously shown that blocking mTOR is useful in alleviating the symptoms of FOP (Hino et al., 2018). What happens to the mTOR pathway in this animal model? How phosphorylated p70S6K and/or pS6 appear?
- 3) In addition, some other therapeutic methods have recently been proposed. The relevance of the results presented needs to be discussed in the discussion section.
- 4) More detailed quantification is required. Ki67 positive cells in Fig1G, how many cells are positive for Ki-67 compared to all cells? DAPI images must be shown and the exact percentage of cells positive for Ki-67 compared to all cells must be given. In addition, Fig2C must be quantified and the panels of d0 must be provided. The same instructions (quantification and d0 images) should be applied to FigS5 to compare the effects.
- 5) In relation to point 4), in Fig1D, they only showed the relative ratio of Ki67 cells to the control condition. The authors must show the exact ratios (percentages) of Ki67-positive cells to total cells. The statistical data must include information on how many panels were analysed and how many cells were counted in each panel. The same applies to Fig2C, where they must provide the exact quantification data.
- 6) statistical analysis; the test used must be clearly stated. In the Methods section, the authors only describe that the statistical analyses were performed using either 2-tailed Student's t-test or 1-way analysis of variance for Dunnett's multiple comparison t-test. The statistical methods used must be described for each test. For example, in Fig1B, if they compared the (-) condition and the BMP-9 condition by Student's t-test, it makes no sense because there is no variation of the data in the (-) condition. The same concern applies to Fig 5G, 5H, 6A, 6D, S1A, S8A, S8D and S9 (comparison between two (-) or control conditions).
- 7) Fig2G; I believe the red circle on the left represents the injured muscle, but I am not sure what the authors meant by the red circle. Also, the figure only illustrates what is currently known about the progression of FOP. In particular, "immune response" is not mentioned in the text or in the data. If the authors try to summarise the data in Fig2, they must include the novelty in the figure. Otherwise, the figure should be withdrawn.
- 8) Fig1G, 3D, 5I, 5J; Some images are black only and the presence of cells cannot be seen. DAPI must be used to ensure that a comparable number of cells are present in the panels being compared.
- 9) line 192; In this section, the authors seem to be using iPS cells to study the molecular mechanisms. This point needs to be clarified as the experimental system has changed from the previous section where they used a mouse model.
- 10) FigS8B: The base of the relative intensity seems to be 0.5. Why? Also, the intensity shown in panel S8A by a western blot looks much more different than shown in (B), where it shows the upregulation by only 2-2.5 times.
(minor)
- 11) The writing needs to be improved. Below are some examples I found in the text, but there are many other places where syntax needs improvement.
 - (a) line 88; They need summary sentences (or paragraphs) at the end of the introduction section.
 - (b) line 100; "notably" - do they mean "however"?
 - (c) line 125; "connection" - do they mean "correlation"?
 - (d) FigS9 figure legend; mRNA was extracted - do they mean "mRNA was analysed"?
- 12) Fig5K,L; It may be useful to indicate what siRNA was used in the figures, not only in the figure legends, to help readers easily understand the experiments performed.

Referee #3 (Comments on Novelty/Model System for Author):

1. Technical Quality

Overall, the result quality is good. However, some of the histological results need to be more accurate.

2. Novelty

Their findings regarding the specific flare-up mediator, BMP9, are somewhat novel. However, previous studies have already shown the importance of TGF- β signaling in FOP pathology. They also applied already established methods, such as FOP-iMSCs.

3. Medical impact

Even if FOP is a rare disease, unmet needs in FOP treatment are innegligible due to its severe symptoms that have huge impacts on patients.

4. Adequacy of model system

As described above, their model system has been well-established and characterized through a series of previous studies.

Referee #3 (Remarks for Author):

In this study, the authors investigated a mechanism that regulates the proliferation of mesenchymal stromal cells (MSCs), leading to flare-ups in FOP. To identify the causal factor that enhances MSC proliferation, the authors utilized MSCs induced from FOP patient-derived iPSCs (FOP-iMSCs). In vitro screening on induced MSCs revealed that BMP9 specifically enhanced proliferation in FOP-iMSCs, but not in MSCs derived from mutation-rescued iPSCs (resFOP-iMSCs). The correlation of BMP9 and flare-ups was also confirmed in FOP model mice. In addition, systemic neutralization of BMP9 in the early phase of the injury and knockout mitigated flare-ups in FOP model mice. This study provided two important findings: FOP-specific flare-up mechanism (BMP9/ACVR1/TGF- β axis) that enhances fibroproliferation of MSCs in the injury sites, and BMP9 as a potential therapeutic target for injury-induced flare-ups in FOP patients.

Their workflow is straightforward, and the strong clinical relevance of this study is worthy of consideration for publication in the journal. However, there are several concerns and points that need to be clarified or revised before publication.

Major concerns:

1. The term, fibroadipogenic progenitors (FAPs), is confusing when used together with MSCs, in terms of its definition (i.e., differentiation capacity into fibroblasts and adipocytes). Or, is there any solid evidence that indicates the differentiation capacity of FAPs into osteo-chondral lineages?
2. In Figure 1G, is there any evidence indicating that the combination of Dox and BMP9 on wild-type mice or a needle insertion into FOP-ACVR1 mice does not cause HO? If not, the authors should show the result of wild-type mice with Dox and BMP9-injection and FOP-ACVR1 mice with only needle insertion as controls. Otherwise, we cannot exclude the above possibilities.
3. In Figure 2B, do the authors have further data obtained later than day 14? It would be crucial to know when the serum BMP9 level peaks out, if the serum BMP9 reflects HO development or progression.
4. In lines 139-140, the authors applied PDGFR α and SP7 as markers for FAPs and osteocytes, respectively, without any references. However, at least, SP7 is not a good osteocyte marker because SP7 is widely expressed in osteogenic lineage and (pre)hypertrophic chondrocytes. PDGFR α is also widely expressed in various cell types. Please cite proper papers and/or consider using other markers.
5. In lines 144-147, the authors concluded that F4/80-expressing monocytes/macrophages during the early stages of inflammation may primarily contribute to FAP proliferation and BMP9 production. However, the authors did not show staining results with PDGFR α and SP7 antibodies at day 3. Thus, they cannot exclude the possibilities that FAPs and osteo-chondral lineages may also express BMP9 at day 3.
6. Figures 2D-2F, please show the whole tissue to indicate the locations of these staining results. Particularly, specific expression of BMP9 would be crucial for the authors' claim.
7. In Figures 3E and 4E, staining results for COL2 and COL1 seem to be obtained from different sections or locations as for HE, SO, and VK. Please use consistent sections.

Minor points:

1. It may be more reader-friendly to spell out abbreviations when they appear first in the main text unless recommended in the journal instruction to omit when abbreviations are spelled out in the abstract: FOP (line 48), HO (line 49), etc.
2. Lines 92-96 are too long to follow. This also disconnects the next sentence (lines 92-96) due to a long phrase, including unnecessarily many citations. Please consider dividing the long sentence into short ones or omitting unnecessary information (i.e., "..., which we developed...").
3. Lines 97-100 are also too long, as mentioned above. In addition, please use consistent phrases unless you intended any special meanings: "Ligands stimulating TGF- β signaling" should be replaced with "Ligands that stimulate TGF- β signaling", like line 99.
4. In lines 101-103, the dose dependency was observed only in the FOP-iMSCs, not resFOP-iMSCs. Thus, resFOP-iMSCs should be removed from this sentence.

5. Please clarify the lines 103-106. It was not clear what the authors intended to mention in the sentence "... and modulated the cell cycle distribution within the G0/G1 phase,"
6. In lines 110-113, please use the consistent expression "day 7" or "d14." Also, please consider reorganizing the figure references: they have been inserted into the middle of the sentences many times, making sentences split.
7. This is related to Major#4. "SP7+ osteocytes" needs to be revised. Are there clear reasons the authors need to stick with osteocytes, not SP7+ osteochondral lineage cells, in which BMP-9 is colocalized?
8. In lines 161-164, the authors describe "COL2-and COL1-positive cells," but in usual contexts, these are supposed to be extracellular matrixes, not intracellular proteins. Thus, this expression seems to be improper.
9. In lines 192-193, the authors used "global gene expression analyses" in this sentence, but used different term, "Transcriptome analysis" in the Method section. Please use the consistent term to describe the same experiments to avoid confusions.
10. In lines 198-200 and Figure 5C, upregulation of TGF-beta pathway seems to be much less as compared to Cell cycle and DNA replication. Because TGF-b signaling via BMP9 is a core mechanism identified in this paper, authors need to elaborate or discuss this difference in the manuscript.
11. In lines 200-202, the term, "discovery" sounds exaggerated. Please consider other words, such as findings and results.
12. In Figures 5I and J, is it possible that even no p-Smad2/3 staining was detected in control groups? Some bands of p-SMAD2/3 were detected in the control group in Figure 5H (although these results were obtained in a much shorter period). In addition, did the authors intentionally change the capitalization of SMAD/Smad between 5H and 5I/J (and line 262)?
13. Lines 229-231 look complicated, and thus should be improved.
14. In lines 237-240, This sentence seems to be incomplete. SB431542 did not suppress BMP9-mediated FOP-iMSCs proliferation. Then, what does this suggest? Did the authors try to mention possible pathways that activate TGF-b signaling other than by the binding of TGF-b and ALK4/5/7? I could not find any interpretation of this result in the manuscript. Please clarify the point of this sentence. Also, please indicate the figure reference properly.

We are very grateful for the reviewer's positive impression and their valuable comments. We have added experiments and carefully revised our manuscript. An item-by-item response to the reviewers' comments is shown below, and the revision was marked in **red** (response to Referee #1), **navy** (response to Referee #2) and **purple** (response to Referee #3) fonts in the manuscript.

***** Reviewer's comments *****

Referee #1 (Comments on Novelty/Model System for Author):

This study employs patient-derived iPSCs and FOP model mice to investigate the role of BMP-9 in mediating fibroproliferation in FOP. The authors used Rosa26::M2rtTA;Col1A1:FOP-ACVR1 mice as the FOP model

Referee #1 (Remarks for Author):

In this manuscript, Zhao et al. demonstrates the involvement of intrinsic BMP-9 in the fibroproliferation of BOP through the abnormal activation of TGF- β signaling. This study employs patient-derived iPSCs and FOP model mice to investigate the role of BMP-9 in mediating fibroproliferation in FOP. Genetic knockout of BMP-9 and administration of BMP-9 neutralizing antibodies significantly mitigate heterotopic ossification, highlighting BMP-9 as a promising therapeutic target for FOP. This provides new insights into potential therapeutic strategies for mitigating HO in FOP. I have a few comments that the authors may take into consideration to improve their manuscript.

Major concerns:

1. The authors suggested that BMP-9 primarily regulated the fibroproliferation of BOP. However, it seems that both chondrogenic and osteogenic capacities were also affected in Bmp9-KO mice (Figure 3E) and in mice treated with the BMP-9 neutralizing antibody (Figure 4E). In addition, the intensity of Safranin O staining was increased in the BMP9 Ab (0-2w) treatment mice. However, the expression of COL2 and COL1 in the BMP9 Ab (1-2w) treatment mice seemed to be minimally affected compared to the control. I think these data need to be quantified, and the authors should examine and explain the impact on chondrogenic and osteogenic capacities.

Thank you for your insightful comments. Regarding the impact of BMP-9 on chondrogenic and osteogenic processes, we propose that BMP-9 primarily promotes the proliferation of MSCs (chondroprogenitors), leading to increased cartilage and bone volume at later stages, rather than directly affecting chondrogenesis and osteogenesis. We have added quantitative analyses to support this view:

1. BMP-9 neutralizing antibody treatment significantly inhibited the proliferation of MSCs in FOP mice (d7, proportion of PDGFR α ⁺/Ki67⁺ cells. **Fig. 4B and C, Results: Lines 201-203**).

2. Regarding the extent of chondrogenesis and osteogenesis:

- After the proliferated MSCs differentiated into chondrocytes which generate ectopic cartilage, the cartilage tissue gradually converts into ectopic bone through endochondral ossification. Therefore, it is quite challenging to quantitatively confirm the extent of cartilage formation, making it difficult to draw a clear conclusion about BMP-9's impact.

- Due to the irregular morphology of ectopic bone, it is experimentally challenging to quantify using representative sections or to digitize data from all serial sections of stained specimens.

For these reasons, we investigated the role of BMP-9 in osteogenesis by analyzing HO tissue using μ CT. BMP-9 inhibition reduced the ectopic bone volume (BV, corresponding to the reduced number of precursor cells forming ectopic bone, Fig. 3C and 4E), while the quality of the bone formed remained consistent (BMD, bone mineral density, Fig. S6 F, S7 C, Results: Lines 185-187, 208-211). This could imply that BMP-9 exerts a minimal influence on the osteogenic process.

2. HO cannot be treated through surgical procedures due to its recurrence at the surgical site. Can Bmp9-KO or the application of Bmp-9 Ab alleviate the recurrence of HO after resection?

Thank you for raising this valuable point. This is a very intriguing hypothesis that merits further exploration. However, simulating the surgical process in mouse model is currently challenging to us due to the technical difficulties in ensuring consistent surgical damage across groups. The recurrence of HO after surgery is primarily due to the muscle tissue damage, similar to the CTX-induced muscle injury model. Based on our results from the CTX-induced FOP model, we hypothesize that Bmp9 knockout or the application of BMP-9 antibody could address the recurrence of HO. We sincerely appreciate your valuable comments regarding this issue. We would like to investigate this issue in further research and have addressed this point in the Discussion section (Discussion: Lines 308-312).

3. In Line 146, the authors stated, "monocytes/macrophages expressing BMP-9 during the early stages of inflammation may contribute to FAP proliferation and BMP-9 production, leading to autocrine expression of BMP9 in FAPs." How do the authors confirm the autocrine expression of BMP9 in FAPs?

Thank you for your comment. In Fig. 2G, we observed that BMP-9 is co-localized with PDGFR α ⁺ FAPs, and DAPI nuclear staining indicates that nearly all cells in this region are PDGFR α ⁺ FAPs. This co-localization strongly suggest that BMP-9 is produced by FAPs themselves. However, as you pointed out, the term "autocrine expression" may not be accurate since we have not directly measured autocrine signaling. Therefore, we have revised the wording to "cell-autonomous expression" (Line 172) to more precisely reflect

our findings. We hope this revision addresses your concerns.

Major concern :

1. The abbreviation "FOP" should not be used in the title.

We acknowledge your suggestion. We have revised the title to include the full term "Fibrodysplasia Ossificans Progressiva" instead of "FOP".

2. The full term should be written out at the first time an abbreviation appears in the main text, even if it has already been defined in the abstract.

Thank you for your suggestion. We have reviewed and updated the relevant sections throughout the entire manuscript (Lines 48, 49, 54, 61, 92, 96 and 104).

3. In line 155, the authors stated, "Similarly, our investigation revealed the viability of adult *Bmp9*-KO; hFOP-ACVR1 (*Bmp9* KO) mice and the normal appearance of their offspring (SI Appendix, Fig. S6 D and E)." However, in Figure 3B, it appears that the femur length of BMP9-KO mice is shortened.

Thank you for your comment. During the reconstruction of the 3D model from microCT data (Figure 3B), to display the entire HO regions and provide a comprehensive view, the femur length of BMP9-KO mice appears shorter due to its orientation perpendicular to the display surface. We performed a quantitative analysis to determine whether there was any difference in tibia and femur length between BMP9-KO and WT mice. The results indicate that the apparent shortening in the figure is likely an artifact of the imaging perspective rather than a true anatomical difference.

4. In Figure 2G, the trends of increased expression of SERPINE1, PMEPA1, and CCL2 in the BMP9 treatment group are inconsistent with the trends shown in the heatmap in Figure 2D.

Thank you for your comment. Indeed, there appears to be a discrepancy between the trends shown in the heatmap (Figure 5D) and the qPCR data (Figure 5G) for *SERPINE1*, *PMEPA1*, and *CCL2* in the BMP9 treatment group. This difference may be attributed to variations in data analysis methods and primer specificity.

To address this, we have converted the RNAseq results (Figure 5D) into a bar graph format similar to Figure 5G. This revised presentation confirms that the overall trends are consistent with BMP-9-specific activation of TGF- β signaling in FOP-iMSCs. Additionally, the expression patterns of other TGF- β signaling-related genes in Figure 5D support this consistency. We believe this aligns with the overall conclusions of our study.

The expression levels of *SERPINE1*, *PMEPA1*, and *CCL2* converted from the heatmap of RNA-seq results (A) and from qPCR data (B).

5. DAPI immunofluorescence needs to provide, e.g. Figures 1G, 3D, 5I, and 5J.

Thank you for your suggestion. We have updated the immunofluorescence images in Figures 1G, 3D, 5I, and 5J to include DAPI staining.

6. The gene and protein names are not consistently formatted.

- "Col1A1" should be written as "Colla1" (Figures 1F, 2A).

- "p-Smad2/3" in Figures 5I and 5J should be "p-SMAD2/3".

Thank you for pointing out these errors. We have revised the formatting of the gene and protein names in both the figures (Figure 1F, 2A, 5I and 5J) and the main text (Lines 39, 244, and 299) accordingly.

Referee #2 (Remarks for Author):

Zhao et al. investigated the molecular mechanisms leading to FOP in soft tissue. Focusing on the early stages of FOP, the authors screened which BMP induces FOP and identified BMP9 as the triggering factor that leads to MSC proliferation and tissue swelling. Using BMP9 KO cells, the authors further demonstrated that endogenous BMP9 is involved in the progression of FOP. The injection of a neutralising antibody against BMP9 supports the idea that blocking BMP9 is effective in inhibiting bursting FOP. RNA sequencing and subsequent experimental analysis (immunohistochemistry and reported assays) further confirmed that BMP9 is a major extracellular molecule that induces MSC proliferation. Finally, the authors showed that BMP9 signal is critically mediated by Smad2/3 to induce the aberrant expression of the TGFb-targeted genes.

Effectively using the combination of animal models and patient-derived iPS cells, the overall study is well conducted and the analyses were thoroughly done. The authors provide a number of compelling data and thus convincingly demonstrate the usefulness of BMP9 in the treatment of FOP. Especially Fig3 and 4 are well-presented and provide hope to patients suffering from this disease.

However, there are several points that I would like the authors to clarify and discuss before making a final decision on publication in the journal.

1) The purpose of the study is to describe how the "painful tissue swelling (flare-ups)" occurs - however, there is no description or data in the results section. If the Fig1G images show the "flare-up" phenotype, the authors must clearly describe and explain this to ensure that the animal model used in this study properly recapitulates the phenotypes of FOP.

Thank you for your insightful feedback. In response to your concerns, we have outlined three major pathological events during flare-ups in the Introduction (Lines 84-86): muscle degeneration, inflammatory cell infiltration, and subsequent fibroproliferative tissue formation. We have emphasized our focus on elucidating the mechanisms of MSCs/FAPs proliferation (Lines 89-91). Additionally, in the Results

section, we have specified that the indicators of flare-ups in FOP model mice include appearance of swelling as the macroscopic indicator and the presence of proliferating FAPs/MSCs (Ki67⁺ PDGFR α -positive cells) as the histological indicator (Lines 121-125). These indicators are based on the fact that swelling is the most common presenting symptom of flare-ups (93%) (Pignolo et al., 2016) and on the study's focus on fibroproliferative tissue formation. We hope these modifications adequately address your concerns.

Pignolo RJ, Bedford-Gay C, Liljestrom M, Durbin-Johnson BP, Shore EM, Rocke DM, Kaplan FS (2016) The Natural History of Flare-Ups in Fibrodysplasia Ossificans Progressiva (FOP): A Comprehensive Global Assessment. *J Bone Miner Res* 31: 650-656

2) The authors' group has previously shown that blocking mTOR is useful in alleviating the symptoms of FOP (Hino et al., 2018). What happens to the mTOR pathway in this animal model? How phosphorylated p70S6K and/or pS6 appear?

Thank you for your interest in the role of mTOR signaling in the pathogenesis of FOP. Due to the difference in focus between the current study (early stage: fibroproliferation) and our previous work on mTOR signaling (later stages: chondrogenesis) in the progression of FOP, this study did not specifically examine the impact of the mTOR pathway. In this study, RNA-seq analysis showed that the mTOR pathway was not included in the gene ontology (GO) pathways significantly regulated by BMP-9 treatment (FOP-iMSCs vs. resFOP-iMSCs, fold change >2, P < 0.05). Additionally, the expression of mTOR pathway-related genes (from the KEGG database) did not show consistent upregulation or downregulation between resFOP/FOP-iMSCs. Therefore, we believe the likelihood of the mTOR pathway playing a role in BMP-9-specific FOP-MS activation is low.

Heatmap illustrating the expression of mTOR pathway-related genes (from the KEGG database) in resFOP/FOP-iMSCs.

Nevertheless, we agree that investigating the impact of the mTOR pathway by assessing phosphorylated p70S6K and/or pS6 in FOP mouse flare-up tissues would be a valuable avenue to explore. Unfortunately, we are not currently housing live FOP model mice for relevant experiments. Reviving frozen embryos and waiting until they reach 20 weeks of age, followed by CTX injection and subsequent staining, would take

nearly five months, which would not allow us to meet the publication timeline. For these reasons, we did not perform above experiments in the current study. We plan to explore the role of this pathway in future research, and we hope this response satisfactorily addresses your concerns.

3) In addition, some **other therapeutic methods** have recently been proposed. The relevance of the results presented needs to be discussed in the **discussion section**.

Thank you for your valuable suggestion. We have discussed the relevance of the presented results and recently proposed therapeutic methods in the discussion section (Lines 316-327, 340-344).

4) More detailed quantification is required. Ki67 positive cells in **Fig1G**, how **many cells are positive for Ki-67** compared to all cells? **DAPI images must be shown** and the exact percentage of cells positive for Ki-67 compared to all cells must be given. In addition, **Fig2C must be quantified and the panels of d0** must be provided. The same instructions (quantification and d0 images) should be applied to **FigS5** to compare the effects.

Thank you for pointing out these issues. We have now included the percentage of Ki-67-positive cells relative to the total cell count in Fig. 1G and provided the corresponding DAPI images (Fig. 1G and H). Additionally, we have quantified the data for Fig. 2C and included the d0 images, as well as the quantification and d0 images for Fig. S5, to allow for a more thorough comparison of the effects (Fig. 2C and D, Fig. S5A and B, and Results: Lines 147-157). We hope these additions address your concerns.

5) In relation to point 4), in Fig1D, they only showed the relative ratio of Ki67 cells to the control condition. The authors must show the exact ratios (percentages) of Ki67-positive cells to total cells. The statistical data must include information on how many panels were analysed and how many cells were counted in each panel. The same applies to Fig2C, where they must provide the exact quantification data.

Thank you for your valuable feedback. In response to your suggestion, we have now included the exact ratios (percentages) of Ki67-positive cells relative to the total number of cells in Fig. 1D. Additionally, we have updated the figure legends of Fig. 1H, Fig. 2C and Fig. S5B, to provide detailed statistical information, including the number of panels analyzed and the number of cells counted per panel. We believe these revisions enhance the clarity and transparency of our data presentation.

6) statistical analysis; the test used must be clearly stated. In the Methods section, the authors only describe

that the statistical analyses were performed using either 2-tailed Student's t-test or 1-way analysis of variance for Dunnett's multiple comparison t-test. The statistical methods used must be described for each test. For example, in Fig1B, if they compared the (-) condition and the BMP-9 condition by Student's t-test, it makes no sense because there is no variation of the data in the (-) condition. The same concern applies to Fig 5G, 5H, 6A, 6D, S1A, S8A, S8D and S9 (comparison between two (-) or control conditions).

Thank you for your valuable feedback regarding the issues related to statistical analysis in our manuscript. We have carefully reviewed and updated the statistical methods for each test. Additionally, we have included a detailed summary of all statistical analyses used in the revised Methods section (Lines 530-538) to ensure clarity and accuracy.

Regarding your observation about the inappropriate comparisons due to the lack of variation in the data for the (-) condition, we sincerely regret this oversight and have reanalyzed the data (Figures 1B, 5G and H, 6A and D, S1, S8B and C, and S9) to better reflect the results for each group and measurement. The figures and statistical methods in the figure legends have been updated accordingly. We hope these revisions adequately address your concerns.

7) Fig2G; I believe the red circle on the left represents the injured muscle, but I am not sure what the authors meant by the red circle. Also, the figure only illustrates what is currently known about the progression of FOP. In particular, "immune response" is not mentioned in the text or in the data. If the authors try to summarise the data in Fig2, they must include the novelty in the figure. Otherwise, the figure should be withdrawn.

Thank you for your valuable comments regarding Fig. 2G. After careful consideration, we agree that the novelty of this figure is limited and have therefore decided to withdraw it from the manuscript. We have chosen to utilize this figure as the synopsis of our manuscript instead.

8) Fig1G, 3D, 5I, 5J; Some images are black only and the presence of cells cannot be seen. DAPI must be used to ensure that a comparable number of cells are present in the panels being compared.

Thank you for your suggestion. We have updated the immunofluorescence images in Figures 1G, 3D, 5I, and 5J to include DAPI staining. However, we would like to note that at the time points shown in the figures (day 7 and day 14), the Dox (+) tissues exhibit fibroproliferation or ectopic cartilage/bone formation, while the Dox (-) tissues are primarily composed of normal muscle cells. As a result, the number of cells within the same field of view may differ. We hope this addresses your concern and appreciate your understanding.

9) line 192; In this section, the authors seem to be using iPS cells to study the molecular mechanisms. This point needs to be clarified as the experimental system has changed from the previous section where they used a mouse model.

Thank you for pointing out this issue. We have clarified the model change in the revised manuscript (Line 224).

10) FigS8B: The base of the relative intensity seems to be 0.5. Why? Also, the intensity shown in panel S8A by a western blot looks much more different than shown in (B), where it shows the upregulation by only 2-2.5 times.

Thank you for pointing out the error in FigS8B. After reviewing the issue, we found that it was due to incorrect axis labeling and failure to subtract the background. We have reanalyzed the data and updated the relevant figure accordingly. We apologize for the oversight and appreciate your understanding.

(minor)

11) The writing needs to be improved. Below are some examples I found in the text, but there are many other places where syntax needs improvement.

(a) line 88; They need summary sentences (or paragraphs) at the end of the introduction section.

(b) line 100; "notably" - do they mean "however"?

(c) line 125; "connection" - do they mean "correlation"?

(d) FigS9 figure legend; mRNA was extracted - do they mean "mRNA was analysed"?

Thank you for pointing out the writing issues in our manuscript. We have carefully reviewed the text and made the necessary revisions. Specifically, we have addressed the following points:

(a) Lines 94-99 (originally line 88): Summary sentences have been added to provide a clearer overview;

(b) Line 112 (originally line 100): The term "notably" has been replaced with "however";

(c) Line 142 (originally line 125): The term "connection" has been replaced with "correlation" for greater accuracy;

(d) FigS9 figure legend: "mRNA was extracted" has been revised to "mRNA was analysed" to accurately reflect the procedure.

We appreciate your feedback and believe these changes have improved the clarity and accuracy of our

manuscript.

12) Fig5K,L; It may be useful to indicate what siRNA was used in the figures, not only in the figure legends, to help readers easily understand the experiments performed.

Thank you for your valuable suggestion. We have included the specific siRNA used in Figures 5K and L within the figures themselves.

Referee #3 (Comments on Novelty/Model System for Author):

1. Technical Quality

Overall, the result quality is good. However, some of the histological results need to be more accurate.

2. Novelty

Their findings regarding the specific flare-up mediator, BMP9, are somewhat novel. However, previous studies have already shown the importance of TGF- β signaling in FOP pathology. They also applied already established methods, such as FOP-iMSCs.

3. Medical impact

Even if FOP is a rare disease, unmet needs in FOP treatment are innegligible due to its severe symptoms that have huge impacts on patients.

4. Adequacy of model system

As described above, their model system has been well-established and characterized through a series of previous studies.

Referee #3 (Remarks for Author):

In this study, the authors investigated a mechanism that regulates the proliferation of mesenchymal stromal cells (MSCs), leading to flare-ups in FOP. To identify the causal factor that enhances MSC proliferation, the authors utilized MSCs induced from FOP patient-derived iPSCs (FOP-iMSCs). In vitro screening on induced MSCs revealed that BMP9 specifically enhanced proliferation in FOP-iMSCs, but not in MSCs derived from mutation-rescued iPSCs (resFOP-iMSCs). The correlation of BMP9 and flare-ups was also confirmed in FOP model mice. In addition, systemic neutralization of BMP9 in the early phase of the injury and knockout mitigated flare-ups in FOP model mice. This study provided two important findings: FOP-specific flare-up

mechanism (BMP9/ACVR1/TGF- β axis) that enhances fibroproliferation of MSCs in the injury sites, and BMP9 as a potential therapeutic target for injury-induced flare-ups in FOP patients.

Their workflow is straightforward, and the strong clinical relevance of this study is worthy of consideration for publication in the journal. However, there are several concerns and points that need to be clarified or revised before publication.

Major concerns:

1. The term, **fibroadipogenic progenitors (FAPs)**, is confusing when used together with MSCs, in terms of its definition (i.e., differentiation capacity into fibroblasts and adipocytes). Or, is there any solid evidence that indicates the differentiation capacity of FAPs into osteo-chondral lineages?

Thank you for pointing out this issue, and we apologize for the lack of clarity in our manuscript. As a subset of MSCs, FAPs were originally identified in muscle tissue and named based on their ability to differentiate into fibroblasts and adipocytes. In recent years, FAPs have been demonstrated to possess the capacity to differentiate into osteo-chondral lineages and are key progenitor cells responsible for ectopic bone formation in muscle tissue, including in FOP (Uezumi *et al*, 2010; Lees-Shepard *et al*, 2018b).

In our manuscript, we used the terms FAPs together with MSCs to refer to the cells driving fibroproliferation, as well as the progenitor cells of ectopic bone formation, to provide readers interested in MSCs or FAPs with a comprehensive understanding of the nature of these cells. We have added clarification regarding this point in the **Introduction section (Lines 53-60)**, and revised the text to address any potentially confusing content in the manuscript.

Uezumi A, Fukada S, Yamamoto N, Takeda S, Tsuchida K (2010) Mesenchymal progenitors distinct from satellite cells contribute to ectopic fat cell formation in skeletal muscle. *Nat Cell Biol* 12: 143-152

Lees-Shepard JB, Yamamoto M, Biswas AA, Stoessel SJ, Nicholas SE, Cogswell CA, Devarakonda PM, Schneider MJ, Jr., Cummins SM, Legendre NP et al (2018b) Activin-dependent signaling in fibro/adipogenic progenitors causes fibrodysplasia ossificans progressiva. *Nat Commun* 9: 471

2. In Figure 1G, is there any evidence indicating that the combination of Dox and BMP9 on wild-type mice or a needle insertion into FOP-ACVR1 mice does not cause HO? If not, the authors should show the result of wild-type mice with Dox and BMP9-injection and FOP-ACVR1 mice with only needle insertion as controls. Otherwise, we cannot exclude the above possibilities.

Thank you for suggesting these important control experiments. We had previously carried out the

experiments you suggested (in the case of FOP-ACVR1 mice (Dox(+)), we performed PBS injection instead of only needle insertion). Results of the analysis at two weeks post-administration indicate no evidence of ectopic bone formation in either wild-type mice receiving Dox and BMP-9 or FOP-ACVR1 mice with Dox and PBS injection (n=5). These results were assessed using X-ray imaging, and since the findings were negative, we did not perform quantitative microCT analysis. We have included these findings in Fig. S4C and the Results section (Lines 129-131). We hope these additions address your concerns.

Wild-type mice Dox (+)

FOP mice Dox (+)

3. In Figure 2B, do the authors have further data obtained **later than day 14**? It would be crucial to know when the **serum BMP9 level** peaks out, if the serum BMP9 reflects HO development or progression.

Thank you for your valuable suggestion. We understand that investigating the serum BMP9 level beyond day 14 would be a valuable avenue to explore, as BMP9 levels could serve as a potential biomarker for HO. Given the persistent presence of ectopic bone tissue in mouse models, we speculate that circulating levels of BMP-9 remain elevated later than day 14.

However, as we selected a relatively shorter timeframe for this study to focus more closely on the dynamic changes in the early stages of FOP, and we are not currently housing live FOP model mice for relevant experiments (reviving frozen embryos and waiting until they reach 20 weeks of age, followed by CTX injection and subsequent analysis, would take nearly five months, which would not allow us to meet the publication timeline), we would like to explore this important aspect in future research. We greatly appreciate your suggestion and will certainly consider extending the observation period in future studies to provide a more comprehensive understanding. Accordingly, we would like to include a discussion regarding this point in the Results section (Lines 159-160).

4. In lines 139-140, the authors applied **PDGF α** and **SP7** as markers for FAPs and osteocytes, respectively, without any references. However, at least, SP7 is not a good osteocyte marker because SP7 is

widely expressed in osteogenic lineage and (pre)hypertrophic chondrocytes. PDGFR α is also widely expressed in various cell types. Please **cite proper papers and/or consider using other markers**.

Thank you for pointing out this important issue. We have revised the manuscript and included appropriate citations when PDGFR α is first mentioned as a marker for FAPs (Lines 125-126). While PDGFR α is indeed expressed in various cell types, it is widely accepted as the most specific single marker for FAPs in muscle tissue, and we have referenced key studies to support this view.

Regarding the use of SP7 as a marker, we acknowledge that SP7 is widely expressed in osteogenic lineages and (pre)hypertrophic chondrocytes (Hojo & Ohba, 2022). Therefore, we have adjusted the wording in the manuscript, modifying the description of SP7⁺ cells to "osteo-chondral lineage cells" (Lines 163-164 and 169), and have cited relevant literature accordingly. We hope this revision clarifies our use of these markers.

Hojo H, Ohba S (2022) Sp7 Action in the Skeleton: Its Mode of Action, Functions, and Relevance to Skeletal Diseases. *Int J Mol Sci* 23

5. In lines 144-147, the authors concluded that F4/80-expressing monocytes/macrophages during the early stages of inflammation may primarily contribute to FAP proliferation and BMP9 production. However, the authors did not show **staining results with PDGFRA and SP7 antibodies at day 3**. Thus, they cannot exclude the possibilities that FAPs and osteo-chondral lineages may also express BMP9 at day 3.

Thank you for raising this important point. In response, we have now included images showing co-staining of BMP-9 and PDGFR α in FOP tissues at day 3 post-CTX injection in Fig. S5D. Additionally, Fig. 2E of the revised manuscript provides BMP-9 immunostaining results. These data suggest that at day 3, F4/80-expressing monocytes/macrophages appear to be the primary source of BMP-9, rather than PDGFR α + FAPs. Given that FAPs typically do not initiate chondrogenesis or osteogenesis until after day 7, SP7 expression was not detected at this earlier time point, which is why related results were not presented. We hope this clarification sufficiently addresses your concerns.

6. Figures 2D-2F, please **show the whole tissue** to indicate the locations of these staining results. Particularly, **specific expression of BMP9** would be crucial for the authors' claim.

Thank you for pointing out this issue. In the revised manuscript, we have now included images of whole tissue BMP-9 immunostaining results at days 3, 7, and 14 post-CTX injection (Fig. 2E), corresponding to Figures 2F-2H (previously Figures 2D-2F). As noted in our response to comment 5, the distribution of BMP-9 is initially concentrated in F4/80-expressing monocytes/macrophages (day 3, with minimal co-expression in FAPs at this stage, Fig. 2F and Fig. S5D). Subsequently, the majority of FAPs begin to express BMP-9 by

day 7 (Fig. 2G). By day 14, in the ectopic cartilage/bone tissue, BMP-9 is primarily localized in the extracellular matrix surrounding osteo-chondral lineage cells (Fig. 2H). We have also added further explanation in the Results section (Lines 161-169) to clarify these observations. We hope these revisions adequately address your concerns.

7. In Figures 3E and 4E, staining results for COL2 and COL1 seem to be obtained from different sections or locations as for HE, SO, and VK. Please use consistent sections.

Thank you for your insightful comment. In the revised manuscript, we have updated the staining images for COL2 and COL1 to improve consistency with the sections used for HE, SO, and VK (Fig. 3E and 4F). We appreciate your careful attention to this detail, as it contributes to the clarity of our findings.

Minor points:

1. It may be more reader-friendly to spell out abbreviations when they appear first in the main text unless recommended in the journal instruction to omit when abbreviations are spelled out in the abstract: FOP (line 48), HO (line 49), etc.

Thank you for your suggestion and for pointing out the specific instances. We have reviewed the entire manuscript and updated the relevant sections to spell out abbreviations when they first appear in the main text (Lines 48, 49, 54, 61, 92, 96 and 104).

2. Lines 92-96 are too long to follow. This also disconnects the next sentence (lines 92-96) due to a long phrase, including unnecessarily many citations. Please consider dividing the long sentence into short ones or omitting unnecessary information (i.e., "..., which we developed...).

Thank you for pointing this out. We have revised the mentioned sentence to improve its readability (Lines 103-106).

3. Lines 97-100 are also too long, as mentioned above. In addition, please use consistent phrases unless you intended any special meanings: "Ligands stimulating TGF-b signaling" should be replaced with "Ligands that stimulate TGF-b signaling", like line 99.

Thank you for your suggestion. We have also revised the mentioned sentence to improve its readability (Lines 108-110).

4. In lines 101-103, the dose dependency was observed only in the FOP-iMSCs, not resFOP-iMSCs. Thus, resFOP-iMSCs should be removed from this sentence.

Thank you for pointing this out. We have revised the mentioned sentence accordingly (Line 114).

5. Please clarify the lines 103-106. It was not clear what the authors intended to mention in the sentence "... and modulated the cell cycle distribution within the G0/G1 phase,"

Thank you for pointing out this issue. We have revised the mentioned sentence "and modulated the cell cycle distribution within the G0/G1 phase" to "and upregulated the cell cycle distribution within the G0/G1 phase" (Line 116).

6. In lines 110-113, please use the consistent expression "day 7" or "d14." Also, please consider reorganizing the figure references: they have been inserted into the middle of the sentences many times, making sentences split.

Thank you for your suggestion. We have revised the manuscript to ensure consistent use of expressions and reorganized the figure references to avoid disrupting the flow of the sentences (Lines 129-131).

7. This is related to Major#4. "SP7⁺ osteocytes" needs to be revised. Are there clear reasons the authors need to stick with osteocytes, not SP7⁺ osteochondral lineage cells, in which BMP-9 is colocalized?

Thank you for your suggestion. In line with your recommendation and consistent with our response to Major #4, we have revised the term "SP7⁺ osteocytes" to "SP7⁺ osteo-chondral lineage cells" (Lines 163 and 169).

8. In lines 161-164, the authors describe "COL2-and COL1-positive cells," but in usual contexts, these are supposed to be extracellular matrixes, not intracellular proteins. Thus, this expression seems to be improper.

Thank you for pointing this out. We have revised the phrase "COL2-and COL1-positive cells" to "COL2-and COL1-positive areas" (Lines 189) to more accurately reflect the extracellular nature of these proteins.

9. In lines 192-193, the authors used "global gene expression analyses" in this sentence, but used different

term, "Transcriptome analysis" in the Method section. Please use the consistent term to describe the same experiments to avoid confusions.

Thank you for pointing this out. We have revised the term "Transcriptome analysis" in the Methods section to "global gene expression analyses" (Line 484) to ensure consistency throughout the manuscript.

10. In lines 198-200 and Figure 5C, upregulation of TGF-beta pathway seems to be much less as compared to Cell cycle and DNA replication. Because TGF-b signaling via BMP9 is a core mechanism identified in this paper, authors need to elaborate or discuss this difference in the manuscript.

Thank you for highlighting this issue. As you noted, the KEGG analysis results in Figure 5C indicate that the changes in the TGF-beta pathway are less pronounced compared to the Cell Cycle and DNA Replication pathways. However, it is important to note that BMP-9 significantly upregulates the TGF-beta pathway in FOP-iMSCs compared to resFOP-iMSCs, with a P -value < 0.001 , as supported by both the KEGG and IPA analyses. Therefore, we believe that the results of the *in silico* analysis are influenced by the algorithms used and should be interpreted as insights rather than absolute conclusions.

We agree that the original presentation could be misleading for readers. To address this, we have made the following adjustments: 1. We modified the color scale in the heatmap of Fig. 5C to better highlight the differences in the TGF-beta pathway; 2. We have included a discussion regarding this difference in the Results section (Lines 231-236). We hope these revisions adequately address your concerns.

11. In lines 200-202, the term, "discovery" sounds exaggerated. Please consider other words, such as findings and results.

Thank you for your suggestion. We have carefully revised this sentence to ensure its accuracy (Lines 235-236).

12. In Figures 5I and J, is it possible that even no p-Smad2/3 staining was detected in control groups? Some bands of p-SMAD2/3 were detected in the control group in Figure 5H (although these results were obtained in a much shorter period). In addition, did the authors intentionally change the capitalization of SMAD/Smad between 5H and 5I/J (and line 262)?

Thank you for highlighting this issue. We appreciate your insight regarding the differences in p-SMAD2/3 detection levels. We believe these differences may arise from the distinct experimental conditions between

in vivo (Figures 5I and J) and *in vitro* (Figure 5H) assays. In Figures 5I and J, we have included DAPI staining, which demonstrates that under the same conditions as the Dox (+) group, no p-SMAD2/3-positive cells were detected in the Dox (-) group. Furthermore, we acknowledge that the sensitivity of western blotting and antibody staining can differ, which may contribute to the observed discrepancies; however, the overall trends in the results are consistent.

Additionally, we apologize for any confusion regarding the capitalization of SMAD/Smad (Figures 5H, I and J, Lines 39, 244, and 299). This was not intentional and has been corrected for consistency.

13. Lines 229-231 look complicated, and thus should be improved.

Thank you for pointing this out. We have made revisions to the sentence you highlighted to enhance its readability (Lines 263-265).

14. In lines 237-240, This sentence seems to be incomplete. SB431542 did not suppress BMP9-mediated FOP-iMSCs proliferation. Then, what does this suggest? Did the authors try to mention possible pathways that activate TGF- β signaling other than by the binding of TGF- β and ALK4/5/7? I could not find any interpretation of this result in the manuscript. Please clarify the point of this sentence. Also, please indicate the figure reference properly.

Thank you for your suggestion, and we apologize for any confusion caused by the insufficient explanation. The observation that SB431542 did not suppress BMP9-mediated FOP-iMSCs proliferation (originally lines 237-240, Fig. 6A), along with the results presented in Fig. 5K, suggests that ACVR1 likely mediates the activation of TGF- β signaling and the proliferation of FOP-iMSCs directly, without involving other Type I receptors. We have clarified this point in the revised manuscript (Lines 275-277). Thank you once again for your thorough review and comprehensive and valuable suggestions.

23rd Oct 2024

Dear Prof. Ikeya,

Thank you for the submission of your revised manuscript to EMBO Molecular Medicine. I am pleased to inform you that we will be able to accept your manuscript pending the following final amendments:

- 1) Please implement referee #2 suggestion.
- 2) In the main manuscript file, please do the following:
 - Please address all comments suggested by our data editors listed below:
 - o Figure legends:
 1. Please note that the exact p values are not provided in the legends of figures 1b, d-e, h; 2b, d; 3c; 4c, e; 5e-h, k-l; 6a-d.
 2. Please indicate the statistical test used for data analysis in the legends of figures 5a, c.
 3. Although 'n' is provided, please describe the nature of entity for 'n' in the legends of figures 1b, e; 3c; 4e; 5e-h, k-l; 6a-d.
 4. Please note that the red arrows are not defined in the legend of figure 4d. This needs to be rectified.
 - Add callouts for Fig 4F.
 - Rename "Conflict of interests" to "Disclosure Statement & Competing Interests". We updated our journal's competing interests policy in January 2022 and request authors to consider both actual and perceived competing interests. Please review the policy <https://www.embopress.org/competing-interests> and update your competing interests if necessary.
 - Indicate in legends exact n and exact p values, not a range, along with the statistical test used. To keep the figures "clear" some authors found providing an Appendix table Sx with all exact p-values preferable. You are welcome to do this if you want to.
 - Please include structured Methods section that includes a Reagents and Tools Table (should be uploaded as a separate file) followed by a Methods and Protocols section. More information on how to adhere to this format as well as downloadable templates (.docx) for the Reagents and Tools Table can be found in our author guidelines: <https://www.embopress.org/page/journal/17574684/authorguide#structuredmethods>
An example of a paper with Structured Methods can be found here: <https://www.embopress.org/doi/full/10.1038/s44320-024-00037-6#sec-4>
 - In Methods, provide the statement that experiments involving human subjects conformed to the principles set out in the WMA Declaration of Helsinki and the Department of Health and Human Services Belmont Report.
 - Author contributions: Please remove it from the manuscript and specify author contributions in our submission system. CRediT has replaced the traditional author contributions section because it offers a systematic machine-readable author contributions format that allows for more effective research assessment. You are encouraged to use the free text boxes beneath each contributing author's name to add specific details on the author's contribution. More information is available in our guide to authors: <https://www.embopress.org/page/journal/17574684/authorguide#authorshipguidelines>
- 3) Appendix: Please rename "Supplementary Information" to "Appendix". Remove the blue font and correct the nomenclature to "Appendix Figure S1" etc and "Appendix Table S1" etc. in the file and in the manuscript text. Add page numbers to the table of contents.
- 4) The Paper Explained: Please provide "The Paper Explained" and add it to the main manuscript text. Please check "Author Guidelines" for more information. <https://www.embopress.org/page/journal/17574684/authorguide#researcharticleguide>
- 5) Synopsis:
 - Synopsis image: Please format the image to 550 px-wide x (300 - 600)-px high and upload it as a high-resolution JPEG file.
 - Please check your synopsis text and image before submission with your revised manuscript. Please be aware that in the proof stage minor corrections only are allowed (e.g., typos).
- 6) As part of the EMBO Publications transparent editorial process initiative (see our Editorial at <http://embomolmed.embopress.org/content/2/9/329>), EMBO Molecular Medicine will publish online a Review Process File (RPF) to accompany accepted manuscripts. This file will be published in conjunction with your paper and will include the anonymous referee reports, your point-by-point response and all pertinent correspondence relating to the manuscript. Let us know whether you agree with the publication of the RPF and as here, if you want to remove or not any figures from it prior to publication. Please note that the Authors checklist will be published at the end of the RPF.
- 7) Please provide a point-by-point letter INCLUDING my comments as well as the reviewer's reports and your detailed responses (as Word file).

I look forward to reading a new revised version of your manuscript as soon as possible.

Yours sincerely,

Zeljko Durdevic

Zeljko Durdevic

*** Instructions to submit your revised manuscript ***

- 1) a .docx formatted version of the manuscript text (including Figure legends and tables)
 - 2) Separate figure files*
 - 3) supplemental information as Expanded View and/or Appendix. Please carefully check the authors guidelines for formatting Expanded view and Appendix figures and tables at <https://www.embopress.org/page/journal/17574684/authorguide#expandedview>
 - 4) a letter INCLUDING the reviewer's reports and your detailed responses to their comments (as Word file).
 - 5) The paper explained: EMBO Molecular Medicine articles are accompanied by a summary of the articles to emphasize the major findings in the paper and their medical implications for the non-specialist reader. Please provide a draft summary of your article highlighting
 - the medical issue you are addressing,
 - the results obtained and
 - their clinical impact.This may be edited to ensure that readers understand the significance and context of the research. Please refer to any of our published articles for an example.
 - 6) Author contributions: the contribution of every author must be detailed in a separate section.
 - 7) EMBO Molecular Medicine now requires a complete author checklist (<https://www.embopress.org/page/journal/17574684/authorguide>) to be submitted with all revised manuscripts. Please use the checklist as guideline for the sort of information we need WITHIN the manuscript. The checklist should only be filled with page numbers where the information can be found. This is particularly important for animal reporting, antibody dilutions (missing) and exact values and n that should be indicated instead of a range.
 - 8) Every published paper now includes a 'Synopsis' to further enhance discoverability. Synopses are displayed on the journal webpage and are freely accessible to all readers. They include a short stand first (maximum of 300 characters, including space) as well as 2-5 one sentence bullet points that summarise the paper. Please write the bullet points to summarise the key NEW findings. They should be designed to be complementary to the abstract - i.e. not repeat the same text. We encourage inclusion of key acronyms and quantitative information (maximum of 30 words / bullet point). Please use the passive voice. Please attach these in a separate file or send them by email, we will incorporate them accordingly.
- You are also welcome to suggest a striking image or visual abstract to illustrate your article. If you do please provide a jpeg file 550 px-wide x 300-600px high.
- 9) A Conflict of Interest statement should be provided in the main text
 - 10) Please note that we now mandate that all corresponding authors list an ORCID digital identifier. This takes <90 seconds to

complete. We encourage all authors to supply an ORCID identifier, which will be linked to their name for unambiguous name identification.

Currently, our records indicate that the ORCID for your account is 0000-0002-3930-8032.

Link Not Available

11) Include a Reagents and Tools Table as part of the Methods section, which can be downloaded from our author guidelines (<https://www.embopress.org/page/journal/17574684/authorguide#structuredmethods>)

Photos 400-800 DPI

*Additional important information regarding figures and illustrations can be found at

<https://bit.ly/EMBOPressFigurePreparationGuideline>. See also figure legend preparation guidelines:

<https://www.embopress.org/page/journal/17574684/authorguide#figureformat>

***** Reviewer's comments *****

Referee #1 (Comments on Novelty/Model System for Author):

This study employs patient-derived iPSCs and FOP model mice to investigate the role of BMP-9 in mediating fibroproliferation in FOP. The authors used Rosa26::M2rtTA;Col1A1:FOP-ACVR1 mice as the FOP model

Referee #1 (Remarks for Author):

My concerns are adequately addressed.

Referee #2 (Comments on Novelty/Model System for Author):

Using both mouse model and human iPS cell is sufficient for proposing a new therapeutic method.

Referee #2 (Remarks for Author):

The authors have sufficiently improved both the writing and the figure formation by considering the opinions of all reviewers. I have just one comment: the DAPI staining in Fig. 2C, S4E,F, and S5A,D appears dim, so it would be better if it could be brightened.

Referee #3 (Remarks for Author):

I thank the authors for revising the manuscript or addressing points according to our comments. Given the journal's revision policy and the quality of this revised manuscript, I think their manuscript is acceptable for publication. I appreciate their contribution to further insights into the FOP mechanisms and identifying a potential therapeutic target for patients.

We are very grateful for the acceptance of our manuscript and for the editor's insights and the reviewer's valuable comments. We have amended our manuscript accordingly. An item-by-item response to the editor's and reviewers' comments is provided below, with revisions marked in red in the manuscript.

*****Comments*****

1) Please implement referee #2 suggestion.

Referee #2 (Remarks for Author):

The authors have sufficiently improved both the writing and the figure formation by considering the opinions of all reviewers. I have just one comment: the DAPI staining in Fig. 2C, S4E,F, and S5A,D appears dim, so it would be better if it could be brightened.

Thank you for your suggestion regarding the DAPI staining in Fig. 2C, S4E, F, and S5A, D. We have adjusted the brightness of the DAPI staining in these figures as recommended to enhance clarity.

2) In the main manuscript file, please do the following: - Please address all comments suggested by our data editors listed below:

o Figure legends:

1. Please note that the exact p values are not provided in the legends of figures 1b, d-e, h; 2b, d; 3c; 4c, e; 5e-h, k-l; 6a-d.

Thank you for pointing this out. We have now included the exact p values in the figure legends for Figures 1b, d-e, h; 2b, d; 3c; 4c, e; 5e-h, k-l; and 6a-d, as requested.

2. Please indicate the statistical test used for data analysis in the legends of figures 5a, c.

The legends for Figures 5a and c have been updated to specify the statistical tests used for data analysis.

3. Although 'n' is provided, please describe the nature of entity for 'n' in the legends of figures 1b, e; 3c; 4e; 5e-h, k-l; 6a-d.

Thank you for your comment. We have now included a description of the nature of the entity for 'n' in the legends of Figures 1b, e; 3c; 4e; 5e-h, k-l; and 6a-d.

4. Please note that the red arrows are not defined in the legend of figure 4d. This needs to be rectified.

Thank you for your feedback. We have rectified this issue by defining the red arrows in the legend of **Figure 4d**.

- Add callouts for Fig 4F.

We apologize for the oversight. Callouts for Figure 4F have now been added to the manuscript in the **Results section (Line 239)**. Thank you for your attention to this detail.

- Rename "Conflict of interests" to "Disclosure Statement & Competing Interests". We updated our journal's competing interests policy in January 2022 and request authors to consider both actual and perceived competing interests. Please review the policy <https://www.embopress.org/competing-interests> and update your competing interests if necessary.

The section has been updated from "Conflict of interests" to "**Disclosure Statement & Competing Interests**". We have also confirmed that the relevant content is accurate. Thank you for your guidance on this matter.

- Indicate in legends exact n and exact p values, not a range, along with the statistical test used. To keep the figures "clear" some authors found providing an Appendix table Sx with all exact p-values preferable. You are welcome to do this if you want to.

Thank you for your feedback. We have provided the exact n and p values, along with the corresponding statistical tests in the figure legends, as addressed in our responses to comments 1 and 3.

- Please include structured Methods section that includes a Reagents and Tools Table (should be uploaded as a separate file) followed by a Methods and Protocols section. More information on how to adhere to this format as well as downloadable templates (.docx) for the Reagents and Tools Table can be found in our author guidelines: <https://www.embopress.org/page/journal/17574684/authorguide#structuredmethods> An example of a paper with Structured Methods can be found here:

<https://www.embopress.org/doi/full/10.1038/s44320-024-00037-6#sec-4>

Thank you for your guidance. We have submitted the Reagents and Tools Table as a separate file, in accordance with the structured Methods section requirements.

- In Methods, provide the statement that experiments involving human subjects conformed to the principles set out in the WMA Declaration of Helsinki and the Department of Health and Human Services Belmont Report.

Thank you for your comment. We have added the requested statement regarding the conformity of our experiments involving human subjects to the principles set out in the WMA Declaration of Helsinki and the Department of Health and Human Services Belmont Report in the Study approval section of the

manuscript (Lines 563-566).

- Author contributions: Please remove it from the manuscript and specify author contributions in our submission system. CRediT has replaced the traditional author contributions section because it offers a systematic machine-readable author contributions format that allows for more effective research assessment. You are encouraged to use the free text boxes beneath each contributing author's name to add specific details on the author's contribution. More information is available in our guide to authors: <https://www.embopress.org/page/journal/17574684/authorguide#authorshipguidelines>

Thank you for your suggestion. We have removed the author contributions section from the manuscript and specified the author contributions in the submission system as recommended.

3) Appendix: Please rename "Supplementary Information" to "Appendix". Remove the blue font and correct the nomenclature to "Appendix Figure S1" etc and "Appendix Table S1" etc. in the file and in the manuscript text. Add page numbers to the table of contents.

Thank you for your guidance. We have made the necessary modifications by renaming "Supplementary Information" to "Appendix", removing the blue font, and correcting the nomenclature to "Appendix Figure S1-9" and "Appendix Table S1-3" in both the file and the manuscript text. Additionally, we have added page numbers to the table of contents in the 1st page of Appendix file.

4) The Paper Explained: Please provide "The Paper Explained" and add it to the main manuscript text. Please check "Author Guidelines" for more information. <https://www.embopress.org/page/journal/17574684/authorguide#researcharticleguide>

Thank you for your feedback. We have added "The Paper Explained" to the manuscript on Page 3 as requested.

5) Synopsis:

- Synopsis image: Please format the image to 550 px-wide x (300 - 600)-px high and upload it as a high-resolution JPEG file.

Thank you for your guidance. We have adjusted the Synopsis image to the specified dimensions of 550 px wide by (300 - 600) px high and have uploaded it as a high-resolution JPEG file.

- Please check your synopsis text and image before submission with your revised manuscript. Please be

aware that in the proof stage minor corrections only are allowed (e.g., typos).

Thank you for the reminder. We have checked the synopsis text and image before submitting the revised manuscript.

6) As part of the EMBO Publications transparent editorial process initiative (see our Editorial at <http://embomolmed.embopress.org/content/2/9/329>), EMBO Molecular Medicine will publish online a Review Process File (RPF) to accompany accepted manuscripts. This file will be published in conjunction with your paper and will include the anonymous referee reports, your point-by-point response and all pertinent correspondence relating to the manuscript. Let us know whether you agree with the publication of the RPF and as here, if you want to remove or not any figures from it prior to publication. Please note that the Authors checklist will be published at the end of the RPF.

Thank you for the information regarding the Review Process File (RPF). We agree to the publication of the RPF and do not wish to remove any figures prior to publication.

31st Oct 2024

Dear Prof. Ikeya,

We are pleased to inform you that your manuscript is accepted for publication and is now being sent to our publisher to be included in the next available issue of EMBO Molecular Medicine.
